# Elephants Never Forget: Testing Language Models for Memorization of Tabular Data

## Abstract

While many have shown how Large Language Models (LLMs) can be applied to a diverse set of tasks, the critical issues of data contamination and memorization are often glossed over. In this work, we address this concern for tabular data. Starting with simple qualitative tests for whether an LLM knows the names and values of features, we introduce a variety of different techniques to assess the degrees of contamination, including statistical tests for conditional distribution modeling and four tests that identify memorization. Our investigation reveals that LLMs are pre-trained on many popular tabular datasets. This exposure can lead to invalid performance evaluation on downstream tasks because the LLMs have, in effect, been fit to the test set. Interestingly, we also identify a regime where the language model reproduces important statistics of the data, but fails to reproduce the dataset verbatim. On these datasets, although seen during training, good performance on downstream tasks might not be due to overfitting. Our findings underscore the need for ensuring data integrity in machine learning tasks with LLMs. To facilitate future research, we release an open-source tool that can perform various tests for memorization `https://github.com/tabmem/tool`.

## 1 Introduction

Large Language Models (LLMs) exhibit remarkable performance on a diverse set of tasks (Wei et al., 2022; Bubeck et al., 2023; Liang et al., 2023). While their prowess in natural language is undeniable, performance in other applications remains an ongoing research topic (Dziri et al., 2023; Nori et al., 2023). A main question in current research on LLMs is the degree to which these models are able to extrapolate to novel tasks that are unlike what they have seen during training (Wu et al., 2023). As such, an important aspect of LLM evaluation is to know to what degree a task might be part of the model's training set and, as a consequence, contained in the LLM's internal representation either verbatim or compressed (Carlini et al., 2021; 2022a).

This paper specifically targets the issue of contamination in training sets when evaluating LLMs on tasks with tabular data — an aspect often neglected in the rapidly growing literature on applications in this domain (Dinh et al., 2022; Borisov et al., 2023; Narayan et al., 2022; Vos et al., 2022; Hegsel-mann et al., 2023; Wang et al., 2023; McMaster et al., 2023). Our investigation reveals that LLMs have been pre-trained on many of the popular datasets, and that this exposure can lead to invalid performance evaluations because the LLMs have, in effect, been fit to the test set. To tackle this issue, we introduce various methods to detect contamination, including statistical tests for conditional distribution modeling, and four different tests to detect memorization (Carlini et al., 2021). To avoid the problem in future research, we release an open-source tool that can perform these tests.

We distinguish between the following three highly related concepts, which can be seen as different types of contamination in the representation learned by an LLM:

- **Knowledge** about a dataset means that the model knows things such as when the dataset was collected, who collected it, the names of the features (column headings), the legal values or ranges of categorical or contunuous variables, the delimiter in the csv file, etc.
- **Learning** from a dataset refers to the model's ability to perform tasks that depend on learning the true joint probability distribution from which the data was sampled, e.g., supervised learning from data to predict a value given other values in a sample.

- **Memorization** of a dataset means that the model knows more about samples in the dataset than could be known by sampling from the true distribution, e.g., it has memorized samples (or parts of samples) perfectly, possibly to all decimal places.

An important contribution of our work is to show that one can make use of the specific structure of tabular datasets in order to distinguish between knowledge, learning, and memorization (Tirumala et al., 2022). This allows us to perform an analysis of how LLMs work with tabular data, in ways that can't easily be replicated with data modalities such as free-form text. An important result of our investigation is to identify a regime where the LLM has seen the data during training and is able to perform complex tasks with the data, but where there is no evidence of memorization.

Our main contributions are the following:

- We emphasize the importance of verifying data contamination before applying LLMs and propose practical methods to do so. We present a variety of tests that allow practitioners to assess whether a given tabular dataset was likely part of the model's training corpus. We also release an open-source tool that can automatically perform the proposed tests.
- We demonstrate the efficacy of the proposed tests on multiple datasets, some of which are publicly available (and thus likely to be in the LLM train sets), as well as on other datasets that have been discussed publicly in papers but where the data itself has not been publicly released. The results demonstrate how the tests are able to distinguish between data that has and has not been seen by LLMs.
- We offer a principled distinction between learning and memorization in LLMs and discuss the implications of data contamination on downstream prediction tasks.

The paper is organized as follows. We specifying the problem in Section 2. In Section 3, we begin our investigation with qualitative tests for what the model knows about the data, introduce the technique of zero-knowledge prompting, and propose a statistical test for whether the model has learned the conditional distribution of the data. In Section 4, we propose four different tests that identify memorization. Section 5 presents the implications of learning and memorization for a downstream prediction task, and Section 6 discusses learning and memorization in LLMs. Section 7 discusses the related work, and Section 8 concludes.

## 2 PROBLEM SETUP

We assume that we have query access to an LLM that we have not trained ourselves. We do not have access to the training data, model architecture, or the model's probability distribution over tokens. We also assume that the model is a chat model, although this assumption is not critical. In short, we assume that we have API access to a proprietary LLM such as GPT-4 or Claude 2.

We would like to evaluate the LLM on a downstream task with tabular data. For example, we might want to study the efficacy of different fine-tuning techniques, optimize the prompts used for in-context learning, or determine the LLMs ability to make accurate statements about the mean, min, max or standard deviation of columns in the data. Because we do not know what data the LLM was trained on, we should be worried that any measured performance might not be due to our applied technique, but because the LLM has already seen the datasets during training that we are using for evaluation. In fact, the LLM might even have memorized the datasets verbatim (Carlini et al., 2021).

**Problem.** How can we test whether an LLM has seen a tabular dataset during it's training? If there is evidence that an LLM has seen a dataset, can we assess the degree of contamination?

**Datasets.** In this study, we use well-known machine learning datasets that are freely avaialable on the Internet, and also less well-known datasets that are from after the cutoff data of the LLMs training, or which have never been made available on the Internet. The publicly available datasets are IRIS, Wine, Kaggle Titanic, OpenML Diabetes (Smith et al., 1988) Adult Income, California Housing (Kelley Pace & Barry, 1997), and the diabetes dataset from scikit-learn (Pedregosa et al., 2011). We also use the FICO HELIOCv1 dataset, which is freely available but guarded by a data usage agreement (Chen et al., 2018); the Kaggle Spaceship Titanic dataset which is from 2022; and Pneumonia, a health care dataset that has never been released on the Internet (Cooper et al., 1997; Caruana et al., 2015). Additional details on the datasets can be found in Supplement A.

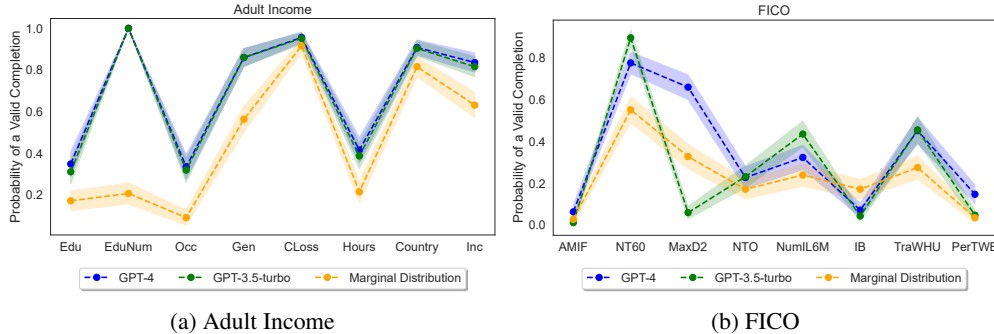

(a) Adult Income             (b) FICO

Figure 1: Conditional completion on Adult Income and FICO. We give the model all previous feature values in the dataset as prefix and ask it to complete the next value in the data. The figure depicts the fraction of correctly completed observations (i.e., an observation exists in the data with that prefix and completion) for three different methods: (1) completion with GPT-3.5, (2) completion with GPT-4, and (3) as a baseline, completion with a random draw from the feature's marginal distribution (i.e., with a random sample from the values of the next column). If GPT-3.5 (green) or GPT-4 (blue) are able to complete rows with true values from the original dataset at a rate higher than the baseline (yellow), this is evidence that the LLMs have seen the data before and memorized parts of it. We show this for 8 different features on 2 datasets. The mean and 95% confidence intervals are shown.

**Language Models.** We conduct all of our experiments with both GPT-3.5 and GPT-4 (Ouyang et al., 2022; OpenAI, 2023).

## 3   TESTING FOR KNOWLEDGE AND LEARNING

In this section, we begin with quick qualitative tests for whether the LLM knows metadata about the dataset such as the names, values and formatting of features. We then study conditional completion to assess if the LLM is able to reproduce statistics of the dataset. We then introduce the technique of *zero-knowledge prompting* and show how it can be used to generate new samples from datasets using only the parametric knowledge of the LLM. This leads us to statistical tests for whether the LLM can accurately model the dataset's conditional distribution.

### 3.1   BASIC METADATA

A simple way to assess a chat model's prior knowledge about a dataset is by asking it. While language models are prone to hallucinations, and may respond confidently with incorrect information, we can focus our questions on metadata that is easy to verify externally (for example, through the use of a datasheet Gebru et al. (2021)). With many popular datasets, it is possible to extract a fair amount of metadata, including the names and values of the features, simply by probing the model in an interactive chat. GPT-4, for example, correctly responds to the question "What possible values can the 'Occupation' feature in the Adult Income dataset take?" and is also able to list exemplary observations from this dataset.

While quick qualitative tests in an interactive chat can be a powerful way to assess a chat models basic knowledge, we have found it important to move beyond this paradigm and design more systematic tests. In particular, we have found it beneficial to systematically condition the model with few-shot examples in order to extract information (Brown et al., 2020). As a concrete example, GPT-4 will claim in an interactive chat that it does not know the names of the features of the FICO dataset, because it has never seen this dataset during training. With proper prompting, we are, however, able to extract the feature names both from GPT-3.5 and GPT-4 (the prompt is depicted in Supplement Figure 6).

The first two columns of Table 1 depict the result of testing for the knowledge of feature names and feature values for all the different datasets considered in this study. GPT-3.5 and GPT-4 know the

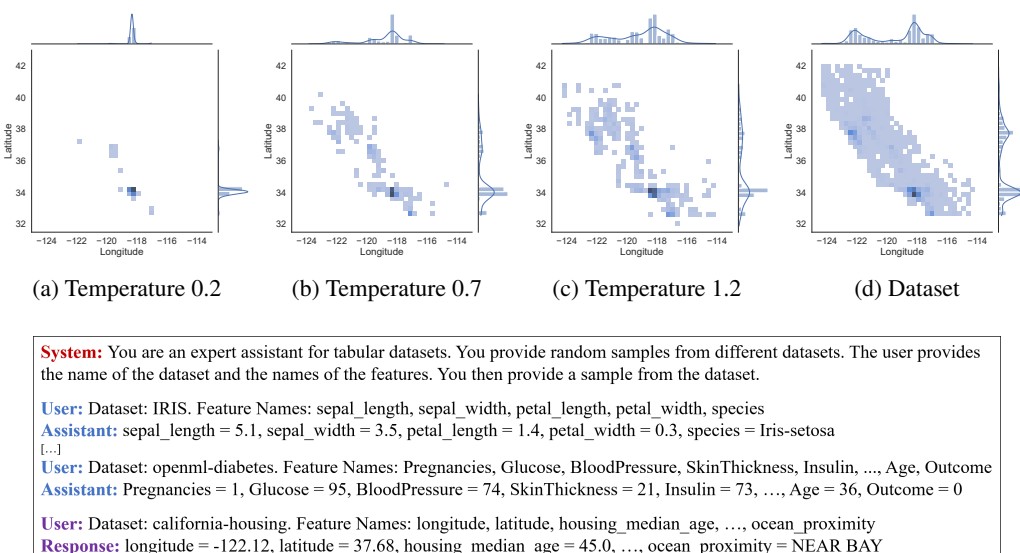

| (a) Temperature 0.2 | (b) Temperature 0.7 | (c) Temperature 1.2 | (d) Dataset |

**System:** You are an expert assistant for tabular datasets. You provide random samples from different datasets. The user provides the name of the dataset and the names of the features. You then provide a sample from the dataset.

**User:** Dataset: IRIS. Feature Names: sepal_length, sepal_width, petal_length, petal_width, species
**Assistant:** sepal_length = 5.1, sepal_width = 3.5, petal_length = 1.4, petal_width = 0.3, species = Iris-setosa
[...]
**User:** Dataset: openml-diabetes. Feature Names: Pregnancies, Glucose, BloodPressure, SkinThickness, Insulin, ..., Age, Outcome
**Assistant:** Pregnancies = 1, Glucose = 95, BloodPressure = 74, SkinThickness = 21, Insulin = 73, …, Age = 36, Outcome = 0

**User:** Dataset: california-housing. Feature Names: longitude, latitude, housing_median_age, …, ocean_proximity
**Response:** longitude = -122.12, latitude = 37.68, housing_median_age = 45.0, …, ocean_proximity = NEAR BAY

Figure 2: Zero-knowledge prompting allows us to draw samples from the LLM's parametric knowledge. Here we ask GPT-3.5 to sample from the California Housing dataset. Zero-knowledge prompting conditions the model on the task without revealing any information about the dataset being tested by providing few-shot examples from *other* datasets. **Top:** The diversity of the generated samples depends on the temperature parameter. For small temperatures, the model produces similar observations concentrated around the mode of data. As temperature increases, the samples become more diverse and more similar to the distribution of the data. At large temperature some samples lie outside the support of the data distribution. **Bottom:** Illustration of the prompt strategy.

feature names and feature values on all the datasets except Spaceship Titanic and Pneumonia.[1] This provides some first evidence that the language models are familiar with these datasets, and might have seen them during training.

## 3.2 TESTING WITH CONDITIONAL COMPLETION

A more systematic way to test a language model's exposure to a dataset is to ask it to complete observations from the dataset. In this approach, we sample a row from the dataset, provide the LLM with the values of the first $n$ features of the row, and then ask it to complete the row as it occurs in the dataset (the prompt is depicted in Supplement Figure 7).

Conditional completion tests if the model is able to reproduce the statistics in the test dataset. However, it is important to note that an LLM might be able to offer a valid completion (a) because it has memorized all the observations in the dataset verbatim, (b) because it has learned the conditional distribution between the different features in the data from the original dataset, or (c) by using transfer learning from other datasets with similar features, or (d) more generally because the LLM is a powerful general-purpose zero-shot predictor.

We believe it is unlikely that current LLMs are able to offer good completions for *all* the different features in a tabular dataset without having seen the dataset during training. The reason for this lies in the particular structure of tabular datasets. Most tabular datasets contain at least some features that are uncommon or possess a fairly specific conditional distribution that would be found in few other places except for the particular dataset. For a concrete example, consider Figure 1, which shows the result of running a conditional completion test for 8 different features both on Adult Income and on FICO. The figure depicts the rate of valid conditional completions for GPT-3.5 and GPT-4, in comparison with the rate of valid completions when drawing from the marginal distribution of the feature. On Adult Income, the LLMs are able to complete all the different features at least as well as the marginal distribution, and some features significantly better (t-test for "EduNum", $p < 0.01$).

---

[1]On Scikit Diabetes, the model possibly responded with standardized feature values.

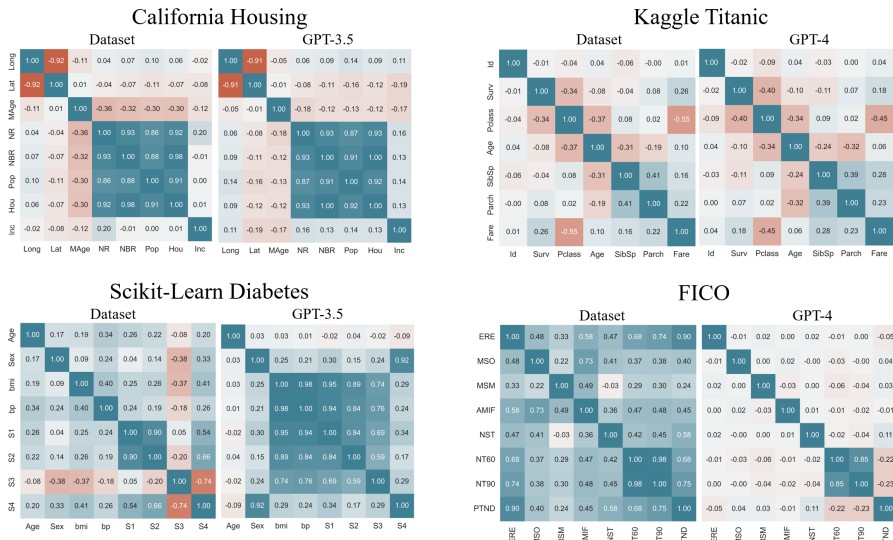

Figure 3: Comparison between the Pearson correlation coefficients in the original dataset and the zero-knowledge samples drawn from GPT-3.5 and GPT-4. For each dataset, we draw 1000 samples with temperature 0.7. Figures for all possible combinations of dataset and language model can be found in the Supplement. The 4 datasets depicted in the Figure were chosen due to their diverse results in Table 1.

On FICO, the LLMs are also able to offer good completions for some of the features. However, there are also features where the completions offered by the language models are significantly worse than draws from the marginal distribution (t-test for "IB", $p < 0.01$), suggesting that the LLMs might have learned very little about these particular features during training.

### 3.3 Testing with unconditional zero-knowledge samples

In the previous section, we developed a simple testing approach using conditional completions. In this section, we introduce another approach to test for learning about a dataset: *unconditional zero-knowledge sampling* of entire observations. *Zero-knowledge prompting*, implicitly already employed in the previous sections, is the solution to the following problem: In order to test the LLM, we cannot provide the details of the dataset in the prompt to prevent in-context learning. At the same time, few-shot learning is required in order to appropriately condition the LLMs to the particular task. The solution is to provide the model with few-shot examples from *other datasets* - in particular, datasets we know the model has seen during training. Figure 2 provides a concrete example of zero-knowledge sampling on California Housing. The top row of Figure 2 compares the distribution of the zero-knowledge samples with the distribution of the same features in the test dataset. It is interesting to see how this depends on the temperature used to sample the observations. For very small temperatures, the samples concentrate around the mode of the data. As we increase the temperature, the samples become more diverse, and for very large temperatures, they exceed the support of the original distribution.

The ability to draw both unconditional (and conditional) samples from a dataset allows us to develop a variety of tests for distribution learning.[2] For example, we can test whether the feature means of the samples are significantly different from the feature means in the original data. This difference is often statistically significant, but not very large (e.g. "Age" on Titanic, feature mean = 30.3, samples mean = 28.8, $p = 0.04$). Instead of providing a battery of statistical tests, Figure 3 presents a visual and intuitive approach: Comparing the Pearson correlation coefficients of the unconditional samples with the correlation coefficients in the original data. Empirically, we find that this serves as a very intuitive test for whether the LLM models the conditional distribution of the data (formally, we could test for whether the signs of the Pearson correlation coefficients in the data and the samples are the

---

[2] We discuss the case when model responses are just copied from the training data in Section 6.

Table 1: Testing for knowledge, learning and memorization of 10 different tabular datasets in GPT-3.5 and GPT-4. The table depicts the result of 8 different tests per dataset where ✓= evidence, ✗= evidence of absence, ?= ambiguous result, and - = test cannot be conducted. For a description of the different tests, see Section 3, Section 4 and Supplement Section C. The Supplement also contains the quantitative test results. The notation */* means result with GPT-3.5/ GPT-4.

| | A. Knowledge and Learning | | | | B. Memorization | | | |
|---|---|---|---|---|---|---|---|---|
| | Feature Names | Feature Values | Feature Distribution | Conditional Distribution | Header Test | Row Compl. Test | Feature Compl. Test | First Token Test |
| Iris | ✓/✓ | ✓/✓ | ✓/✓ | ✓/✓ | ✓/✓ | ?/✓ | -/- | ?/✓ |
| Wine | ✓/✓ | ✓/✓ | ✓/✓ | ?/? | ✓/✓ | ?/✓ | ✓/✓ | -/- |
| Kaggle Titanic | ✓/✓ | ✓/✓ | ✓/✓ | ✓/✓ | ✓/✓ | ✓/✓ | ✓/✓ | -/- |
| OpenML Diabetes | ✓/✓ | ✓/✓ | ✓/✓ | ✓/✓ | ✓/✓ | ?/✓ | ✓/✓ | ✗/✓ |
| Adult Income | ✓/✓ | ✓/✓ | ✓/✓ | ✓/✓ | ✓/✓ | ✗/✗ | ✗/✗ | ✗/✗ |
| California Housing | ✓/✓ | ✓/✓ | ✓/✓ | ✓/✓ | ✓/✓ | ✗/✗ | ✗/✗ | -/- |
| Scikit Diabetes | ✓/✓ | ?/? | -/- | -/- | ✓/✓ | ✗/✗ | ✗/✗ | ✗/✗ |
| FICO | ✓/✓ | ✓/✓ | ?/? | ✗/✗ | ✗/✗ | ✗/✗ | ✗/✗ | ✗/✗ |
| Spaceship Titanic | ✗/✗ | ✗/✗ | -/- | -/- | ✗/✗ | ✗/✗ | ✗/✗ | -/- |
| Pneumonia | ✗/✗ | ✗/✗ | -/- | -/- | ✗/✗ | ✗/✗ | ✗/✗ | ✗/✗ |

same). On California Housing and on Kaggle Titanic, depicted in the top row of Figure 3, the model provides samples that match the Pearson correlations in the original data remarkably well. On Scikit Diabetes and FICO, depicted in the bottom row of Figure 3, this is not the case. On FICO, where the model knows the names of the features and is able to provide correctly formatted observations, the LLM's inability to accurately replicate the feature correlations suggests that it may not have memorized (or even seen) the entire data during its training.

## 4 TESTING FOR MEMORIZATION

In the previous section, we discussed different strategies to test for knowledge and learning. In this section, we introduce four different tests to detect the memorization. In particular, our goal now is to provide tests that definitively distinguish memorization from learning (Carlini et al., 2021; 2022b; Chang et al., 2023). All tests rely on assumptions. One assumption is that the dataset has non-zero entropy (i.e., the dataset is not comprised of many copies of the same row). For some of the proposed tests, we also assume that there exists a canonical csv file of the dataset, that the rows in this csv file are ordered at random, or that there exists a feature in the dataset with highly unique values.

It is useful to think of the proposed tests in terms of power and significance. By design, all the proposed tests are highly significant. This means that any of the tests being positive is strong evidence of memorization (though not necessarily of the entire dataset). However, the power of the tests is difficult to estimate because the tests rely on prompting and it is possible that the LLM has memorized the data but we cannot extract it via prompting. The proposed tests are as follows:

1. **Header Test.** We provide the model with the first couple of rows of the csv file of the dataset and ask it to complete the next rows, as they occur in the csv file (that is, we ask the model to complete the 'header' of the dataset). We use zero-knowledge prompting to condition a chat model on this task.

2. **Row Completion Test.** We provide the model with a number of contiguous rows from a random position of the csv file of the dataset and ask it to *perfectly* complete the next row in the file. If the rows of the dataset are known to be unordered, we can perform a $t$-test between the similarity of model completions with actual vs. random rows (Navarro, 2001).

3. **Feature Completion Test.** If there exists a feature that takes unique values, or almost unique values, we ask the model to *perfectly* complete the value of this unique feature, given all the other features of an observation. Examples of unique features are names and numbers with many decimal places.

4. **First Token Test.** We provide the LLM with a number of contiguous rows from a random position of the csv file and ask it to complete the first token of the next row in the csv. If the rows of the dataset are known to be random, we compare the accuracy of the completions to the accuracy of completion with the mode. If the rows are non-random, we heuristically compare the accuracy of the completions to the test accuracy of a classifier that, given the values of the observations in the previous rows, predicts the first token of the next row.

Table 1 shows the results of the four tests on 10 tabular datasets. For a detailed description of the datasets, see Supplement A. Table 1 shows the overall results of the tests, i.e., whether a test provides evidence for memorization or not. For the detailed quantitative test results, see Supplement C. The *header test* succeeds for all datasets that are publicly available on the internet. This means that GPT-3.5 and GPT-4 have memorized the initial rows of these datasets verbatim. On four publicly available datasets (Iris, Wine, Kaggle Titanic, OpenML Diabetes) the *row completion* and *feature completion* tests also largely succeed. This means that the LLMs have not only memorized the initial rows, but random rows from the dataset. The same holds true for the *first token test*. We observe that the results of the latter three test are overall consistent, with a tendency of GPT-4 to exhibit more memorization than GPT-3.5. Interestingly, there are 3 datasets (Adult, Housing, Scikit Diabetes) where the header tests succeeds, but all the other memorization tests fail. This means that the LLMs have memorized the initial rows of these datasets, but not memorized random rows.[3] Finally, the memorization tests give no evidence of memorization on FICO, Spaceship Titanic and Pneumonia. On Spaceship Titanic and Pneumonia this is expected. The result on FICO suggests that this dataset might have been protected from inclusion in the LLM training data by the need to register prior to access. (Of course, it is possible that the LLM saw the dataset in training, but there was no memorization, or that our memorization tests are not sensitive enough.)

## 5 IMPLICATIONS FOR A DOWNSTREAM PREDICTION TASK

In the previous sections we developed different tests for learning and memorization, and discussed to what degree GPT-3.5 and GPT-4 have memorized popular tabular datasets. In this section we study the consequences for a downstream prediction task. We consider the task of using the LLM for prediction with few-shot learning. We encode the observations of the tabular dataset in textual form, provide the model with 20 randomly selected few-shot examples and a test case, ask it predict the label, and measure the predictive accuracy. For prompt details, see Supplement Figure 12.

Table 2: Few-shot binary classification with large language models. The table depicts the predictive accuracy of GPT-4, GPT-3.5, gradient boosted trees and logistic regression on 6 tabular datasets. The language models use 20 randomly selected few-shot examples. The traditional learning algorithms are trained on the entire dataset (excluding the test set). Leave-one-out cross validation is used for small datasets. 95% confidence intervals in parenthesis.

|  | Kaggle Titanic | OpenML Diabetes | Adult Income | FICO | Spaceship Titanic | Pneumonia |
|---|---|---|---|---|---|---|
| GPT-4 | **0.98** (0.97, 0.99) | 0.75 (0.72, 0.78) | 0.82 (0.80, 0.85) | 0.68 (0.65, 0.71) | 0.69 (0.66, 0.72) | 0.81 (0.79, 0.83) |
| GPT-3.5 | 0.82 (0.80, 0.85) | 0.74 (0.70, 0.77) | 0.79 (0.76, 0.81) | 0.65 (0.62, 0.68) | 0.63 (0.59, 0.66) | 0.54 (0.50, 0.56) |
| Gradient Boosted Tree | 0.84 (0.81, 0.86) | 0.75 (0.72, 0.78) | **0.87** (0.87, 0.88) | **0.72** (0.70, 0.74) | **0.80** (0.79, 0.82) | **0.90** (0.89, 0.91) |
| Logistic Regression | 0.79 (0.76, 0.82) | **0.78** (0.74, 0.80) | 0.85 (0.85, 0.86) | **0.72** (0.70, 0.74) | 0.77 (0.74, 0.80) | **0.90** (0.89, 0.91) |

---

[3]Why is this the case? We offer the following hypothesis: Jupyter Notebooks on Kaggle and other platforms frequently employ the pandas function 'head()' to print the initial rows of a datasets. As a consequence, web search for the initial rows of datasets often leads to results, whereas web search for random rows does not. This likely means that the LLMs see the initial rows more often during training than random rows. The result would then be consistent with the observation in previous works that seeing a string more often during training increases the chance of that string being memorized by the model (Carlini et al., 2022b).

Table 3: Summary statistics of the zero-knowledge samples for temperature 0.7 (compare Figure 2, Figure 3, and the Figures in Supplement D).

|  | OpenML Diabetes | | Adult Income | | California Housing | |
|---|---|---|---|---|---|---|
|  | GPT-3.5 | GPT-4 | GPT-3.5 | GPT-4 | GPT-3.5 | GPT-4 |
| Copied observations | 52.7% | 53.3% | 0.0% | 11.4% | 0.1% | 10.4% |
| Best n-gram match | 7.8/9 | 7.3 / 9 | 13.3 / 15 | 13.7 / 15 | 3.1 / 10 | 4.4 / 10 |
| Copied feature values | 99.90% | 99.94% | 99.74% | 99.79% | 99.53% | 99.64% |

Table 2 shows the test accuracy of GPT-4, GPT-3.5, a Gradient Boosted Tree and Logistic Regression for 6 datasets from Table 1. On Kaggle Titanic and OpenML Diabetes, two publicly available dataset that are highly memorized (Table 1), GPT-4 and GPT-3.5 perform on-par with the traditional learning algorithms. On Kaggle Titanic, the 'performance' of GPT-4 at 98% test accuracy can safely be attributed to memorization. Moreover, GPT-4 and GPT-3.5 also show good performance on Adult and FICO. The result on Adult is unsurprising since the LLMs have learned the conditional distribution of the data (Figure 1). The case of FICO is more interesting, since there is little evidence that the LLMs have learned or memorized this dataset during training. It might be that this learning task is relatively simple, that our memorization test are not sensitive enough, or that FICO constitutes a genuine case of generalization towards an unseen dataset. On Spaceship Titanic and Pneumonia – the two datasets for which there is no evidence that the LLMs saw them during training (compare Table 1) – we see a marked drop in predictive accuracy, especially for GPT-3.5. If we compare the results on Titanic and Spaceship Titanic, we see that traditional learning algorithms perform comparably to each other whereas the performance of the LLMs drops significantly. On Pneumonia, the test accuracy of GPT-3.5 is no better than random guessing.

Interestingly, we observe that GPT-4 outperforms GPT-3.5 on FICO, Spaceship Titanic and Pneumonia, suggesting that there are differences in the ability of these two models to generalize to novel tasks. Nevertheless, GPT-4 performs significantly better on seen data, suggesting that *the model that has the best ability to memorize might also be the model that has the best ability to generalize to novel tasks.*

## 6    LEARNING AND MEMORIZATION

In this section, we investigate the structure of the zero-knowledge samples in more detail. This will shed some light on learning and memorization in LLMs. Table 3 depicts summary statistics of the zero-knowledge samples of GPT-3.5 and GPT-4. The first row depicts the fraction of the samples that are copied from the training data. The second row depicts the best n-gram match between a sample and the training data. An n-gram match of 7.8 out of 9 means that the closest match of a sample in the training data shares, on average, 7.8 out of 9 feature values. The third row depicts the fraction of the values of the individual features of the samples that also occur in the training data.

On OpenML diabetes (first two columns of Table 3), more than 50% of the 'samples' generated by the LLMs are copied from the dataset. Unsurprisingly, this leads to a very high n-gram match and a very large fraction of copied feature values from the training data. On Adult Income, interestingly, GPT-3.5 copies zero exact rows from the training data but still has a very high n-gram match of 13.3/15. Why is this? The cause is the feature `fnlwgt` which is the weight of an observation in the US Census data that the dataset is based on. The values of this feature are highly unique, and largely uncorrelated with the remaining features and LLMs consistently fail to reproduce this feature. We also note that while the relatively high value of 13.3/15 might suggest memorization, this statistic might be seen as a reflection of the low entropy of the training data (and hence, as learning). On Adult Income, the best n-gram match of an average row with the other rows in the dataset is 12.7/15.

The California Housing samples are much more diverse than Adult Income: the average GPT-3.5-sample shares only 3.1 out of 10 feature values with the training data. This is again a reflection of the statistics in the training data, where the ratio is 3.07/10. At the same time, on all the datasets, the LLM only produces feature values that are also present in the training data. This provides an interesting perspective on learning and memorization in LLMs. While the LLM on California Housing has apparently *learned* to build new plausible combinations of the features (the Pearson

correlation coefficients of the generated data match the statistics of the training data in Figure 3), the LLM still copies the values of the *individual* features directly from the training data.[4]

# 7 RELATED WORK

Many recent works have demonstrated the capabilities of LLMs on tasks with tabular data (Dinh et al., 2022; Narayan et al., 2022; Vos et al., 2022; Wang et al., 2023; McMaster et al., 2023). In particular, Borisov et al. (2023) and Hegselmann et al. (2023) have shown that LLMs can be fine-tuned to generate and classify tabular data. A consistent gap in the existing literature on LLMs for tabular data is that none of the aforementioned works attempt to test the LLM for memorization. At the same time, there is increasing interest in the topic of memorization, especially in LLMs (Carlini et al., 2021; 2022b). The topic has been studied in a variety of different contexts, for example, copyright infringement (Liang et al., 2022), and also for diffusion models (Somepalli et al., 2023). There has also been an increasing interest in the relationship between learning and memorization (Tirumala et al., 2022), and the literature on membership inference attacks asks how one can detect if an observation was part of the training data (Carlini et al., 2022a; Mattern et al., 2023). This literature assumes access to the probability distribution over tokens, or the ability to re-train the model, whereas we assume only blackbox API access. Our header- and row completion tests work are closely related to the memorization tests developed in (Carlini et al., 2021). The feature completion test is similar to the approach taken in Chang et al. (2023). Nori et al. (2023) also takes a blackbox testing approach for memorization of medical exam questions, but focuses on the text domain.

# 8 DISCUSSION

In this work, we introduced three separate but highly related notions of representation learning – "knowledge", "learning", and "memorization" – and developed various tests for assessing a model on these dimensions. We have seen that some publicly available datasets are highly memorized, meaning that LLMs can reproduce these datasets verbatim (Section 4). We have also seen that this can lead to invalid performance estimates on downstream tasks (Section 5). While some datasets are fully memorized, others are only partially memorized. For example, an LLM may have memorized the initial rows of some datasets but failed to recognize the rest of the rows (Section 4). That said, even when there is only partial memorization, we find evidence that LLMs have still learned the conditional distribution between features, suggesting that the data might be internal to the LLM in compressed form (Section 3).

Although we study the specific domain of tabular data, we find similar patterns with respect to memorization as previous works that analyzed different domains (Carlini et al., 2022b). In particular, we find that (1) the initial rows of datasets are more likely to be memorized, (2) larger LLMs consistently memorize more, and (3) it is important to use zero-shot prompting in order to condition the chat models on the task.

An interesting insight of our work is that we are able to condition the model to provide samples without revealing any details of the dataset in question via zero-knowledge prompting. We then analyze the structure of these samples in order to assess the relationship between learning and memorization (Section 6). This approach might have broader application beyond tabular data.

A limitation of our work is that we do not have access to the training data of GPT-3.5 and GPT-4. For example, we do not know if the training data of GPT-3.5 is a subset of the training data of GPT-4 or not. As such, we can draw conjectures such as the hypothesis that GPT-4 is "better" at memorization because it is bigger, but ultimately these must remain conjectures. Additionally, the experiments are tied to the capabilities of current LLMs. For example, GPT-3.5 and GPT-4 are able to provide well-formatted observations from California Housing and Adult Income even if the order of the features is permuted (required by our feature completion test). With weaker LLMs this test might not be possible.

---

[4]We also observe that GPT-4 repeatedly copies observations from the training data of California Housing and Adult Income. It turns out that these are largely repeated copies of the first 10 rows of the dataset (the 'header'), which the model has memorized (compare Table 1).

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

## A   DATASETS

**Iris.** The Iris flower dataset is a small dataset (150 observations) that is freely available on the Internet, among others in the UCI repository at `https://archive.ics.uci.edu/dataset/53/iris`.

**Wine.** The UCI Wine dataset is another small dataset from the UCI repository (178 observations). It is available at `https://archive.ics.uci.edu/ml/machine-learning-databases/wine/wine.data`.

**Kaggle Titanic.** The Kaggle Titanic dataset is a very popular freely available machine learning dataset `https://www.kaggle.com/competitions/titanic`. Every observation in the dataset starts with a unique ID. The dataset also contains the unique feature "Name".

**OpenML Diabetes.** The OpenML Diabetes dataset (Smith et al., 1988) is a popular dataset that is freely available `https://www.openml.org/search?type=data&sort=runs&id=37&status=active` as part of OpenML (Vanschoren et al., 2014).

**Adult Income.** Historically, the Adult Income dataset is one of the most popular machine learning datasets `http://www.cs.toronto.edu/~delve/data/adult/adultDetail.html`. Recently, researchers have argued that this dataset should no longer be used Ding et al. (2021). The csv file of the dataset can still be found at Kaggle `https://www.kaggle.com/datasets/wenruliu/adult-income-dataset` and at many other places. Apart from the fact that it is replicated many times over the Internet, the Adult Income dataset has some properties that make it ideal for our experiments: (1) the ordering of the rows in the csv file is known to be random, (2) the dataset contains the (almost) unique feature `fnlwgt`, and (3) the dataset is not small, consisting of 32561 training observations and an additional 16281 test observations.

**California Housing.** The California Housing dataset (Kelley Pace & Barry, 1997) is a freely available and very popular machine learning dataset with 20640 observations.

**Scikit Diabetes.** The Scikit Diabetes dataset is available via the popular scikit-learn python library `https://scikit-learn.org/stable/datasets/toy_dataset.html#diabetes-dataset`. We run our experiments with the dataset as it is available via the function `load_diabetes` in scikit-learn. We note that there exists another version of the dataset, available at `https://www4.stat.ncsu.edu/~boos/var.select/diabetes.rwrite1.txt`.

**FICO.** The FICO HELIOCv1 dataset was part of the FICO Explainable Machine Learning Challenge `https://community.fico.com/s/explainable-machine-learning-challenge`. This dataset can be obtained only after signing a data usage agreement and is then available via Google Drive. This is an example of a dataset that is freely available, but where the canonical csv file has not been publicly released on the internet, at least not by the creators of the dataset.

**Spaceship Titanic.** The Spaceship Titanic dataset is available from Kaggle at `https://www.kaggle.com/c/spaceship-titanic`. This dataset is part of a Kaggle competition from 2022. This means that the dataset was made available only after the training data cut-off date of GPT-3.5 and GPT-4.

**Pneumonia.** The pneumonia dataset was first used in Cooper et al. (1997) and has since been employed by a variety of researchers. The dataset has never been publicly released on the Internet.

## B   MODELS

**Language Models.** The experiments were performed with `gpt-4-32k-0314`, `gpt-4-0613`, `gpt-3.5-turbo-16k-0613` and `gpt-3.5-turbo-0301`.

**Logistic Regression.** We train logistic regression using scikit-learn (Pedregosa et al., 2011). We cross-validate the $L_2$ regularization constant.

**Gradient Boosted Tree.** We train gradient boosted trees using xgboost (Chen & Guestrin, 2016). We cross-validate the max_depth of the trees and the $L_1$ and $L_2$ regularization constants, similar to (Hegselmann et al., 2023).

## C  TESTS

This section gives additional details on the different tests, as well as the quantitative results of the memorization tests. As a general observation, one might expect the results of the tests to be somewhat ambiguous, especially the quantitative memorization tests that rely on assumptions. Instead, the results of the different tests turned out to be surprisingly clear-cut, with only very few ambiguous cases (and these are mostly small datasets with little entropy, where it can be difficult to distinguish memorization from prediction).

**Zero-Knowledge Sampling.** In the tests that use zero-knowledge sampling, we use the following few-shot datasets to condition the model responses: Iris, Adult Income, Kaggle Titanic, Wine, and California Housing. When we are testing one of these five datasets, we use OpenML Diabetes for the few-shot examples instead.

### C.1  FEAURE NAMES

The feature names test was conducted using the prompt structure given in Figure 6. In all cases, the result of this test was unambiguous. The model would either list the names of the different features in exactly the same format as it occurs in the csv file of the dataset, or it would offer a completely different response.

### C.2  FEATURE VALUES

To test whether the model is able to model the feature values of the data, we draw unconditional zero-knowledge samples. We then test whether the formatting of the sampled feature values is the same as the formatting in the training data (similar to the evaluation in the third row of Table 3 in the main paper). For moderate temperature levels, there is no significant effect of the sampling temperature on the test results.

### C.3  FEATURE DISTRIBUTION

We use unconditional zero knowledge samples for a small temperature level and compare the mode of the sampled feature values with the mode of the feature in the dataset. The motivation behind this comparison is that an autoregressive model that goes over the data sequentially and knows the ordering of the different features but has not learned any aspect of the conditional distribution between the different features, should still assign the highest likelihood to the modal value of the feature (by definition, the mode is the unconditional prediction with the best accuracy). This test is also motivated by the empirical observation that GPT-3.5 and GPT-4 model the modal feature values of many publicly available datasets surprisingly well, even if they rarely provide a good model of the finer details of the feature distribution. The test was unambiguous on all datasets except FICO. The test can only be conducted if the feature values test succeeds.

### C.4  CONDITIONAL DISTRIBUTION

We use unconditional zero knowledge samples with temperature 0.7 and compare the Pearson correlation coefficients of the samples with the Pearson correlation coefficients in the original dataset (see Figure 4 and Figure 5).

### C.5  HEADER TEST

We split the dataset at random positions in rows 2, 4, 6, and 8 and ask the model to complete the dataset from the given position. We condition the model on this task with few-shot examples and consider the best completion. The test result was unambiguous on all datasets (either the model completed many rows, or not even a single row). The prompt is depicted in Figure 9.

## C.6 ROW COMPLETION, FEATURE COMPLETION, AND FIRST TOKEN TEST

Table 4: Detailed results of the row completion and feature completion tests from Table 1 in the main paper. For each dataset and each model, the table depicts the number of rows and features that were correctly completed. The prompts are depicted in Figure 10 and Figure 11.

|  | Row Completion Test | | Feature Completion Test | | |
|---|---|---|---|---|---|
|  | GPT-3.5 | GPT-4 | GPT-3.5 | GPT-4 | Feature Name |
| Iris | 35 / 136 | 125 / 136 | - | - | - |
| Wine | 16 / 164 | 84 / 164 | 77 / 178 | 131 / 178 | malic_acid |
| Kaggle Titanic | 194 / 250 | 222 / 250 | 238 / 250 | 236 / 250 | Name |
| OpenML Diabetes | 18 / 250 | 79 / 250 | 237 / 250 | 243 / 250 | DiabetesPedigreeFunction |
| Adult Income | 0 / 250 | 0 / 250 | 0 / 250 | 0 / 250 | fnlwgt |
| California Housing | 0 / 250 | 0 / 250 | 0 / 250 | 1 / 250 | median_income |
| Scikit Diabetes | 0 / 250 | 0 / 250 | 1 / 250 | 1 / 250 | S2 |
| FICO | 1 / 250 | 2 / 250 | 2 / 250 | 14 / 250 | MSinceOldestTradeOpen |
| Spaceship Titanic | 0 / 250 | 0 / 250 | 0 / 250 | 2 / 250 | Name |
| Pneumonia | 0 / 250 | 0 / 50 |  | 1 / 250 | WBC Count |

Table 5: Detailed results of the first token test from Table 1 in the main paper. The table depicts the number of first tokens that were correctly completed and the overall accuracy of the completions (in brackets). The last row depicts the completion accuracy that can be reached with traditional predictors (that is, without memorization). The prompt is depicted in Figure 10.

|  | First Token Test | | |
|---|---|---|---|
|  | GPT-3.5 | GPT-4 | Baseline Accuracy (Best of Mode, LR and GBT) |
| Iris | 88 / 136 (0.65) | 131 / 136 (0.96) | 0.50 |
| Wine | - | - | 0.95 |
| Kaggle Titanic | - | - | - |
| OpenML Diabetes | 42 / 250 (0.17) | 95 / 250 (0.38) | 0.25 |
| Adult Income | 59 / 250 (0.24) | 68 / 250 (0.27) | 0.26 |
| California Housing | - | - | 0.95 |
| Scikit Diabetes | 66 / 250 (0.26) | 54 / 250 (0.22) | 0.40 |
| FICO | 119 / 250 (0.48) | 78 / 250 (0.31) | 0.47 |
| Spaceship Titanic | - | - | - |
| Pneumonia | 0 / 250 (0.0) | 11 / 50 (0.22) | 0.26 |

# D    ADDITIONAL FIGURES

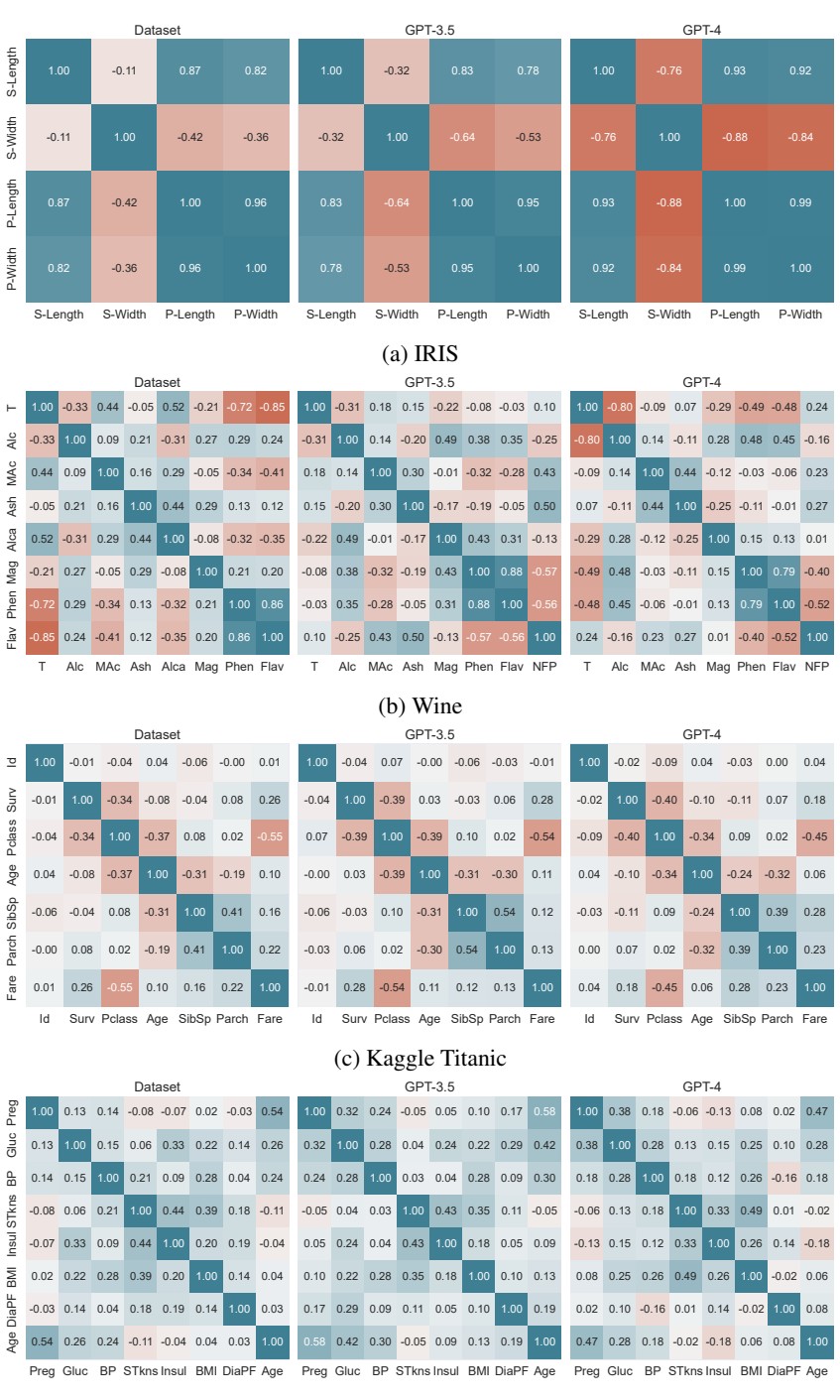

Figure 4: Pearson Correlation coefficients for all possible combinations of dataset and language model. Compare Figure 3 in the main paper. Temperature 0.7. Continued on next page.

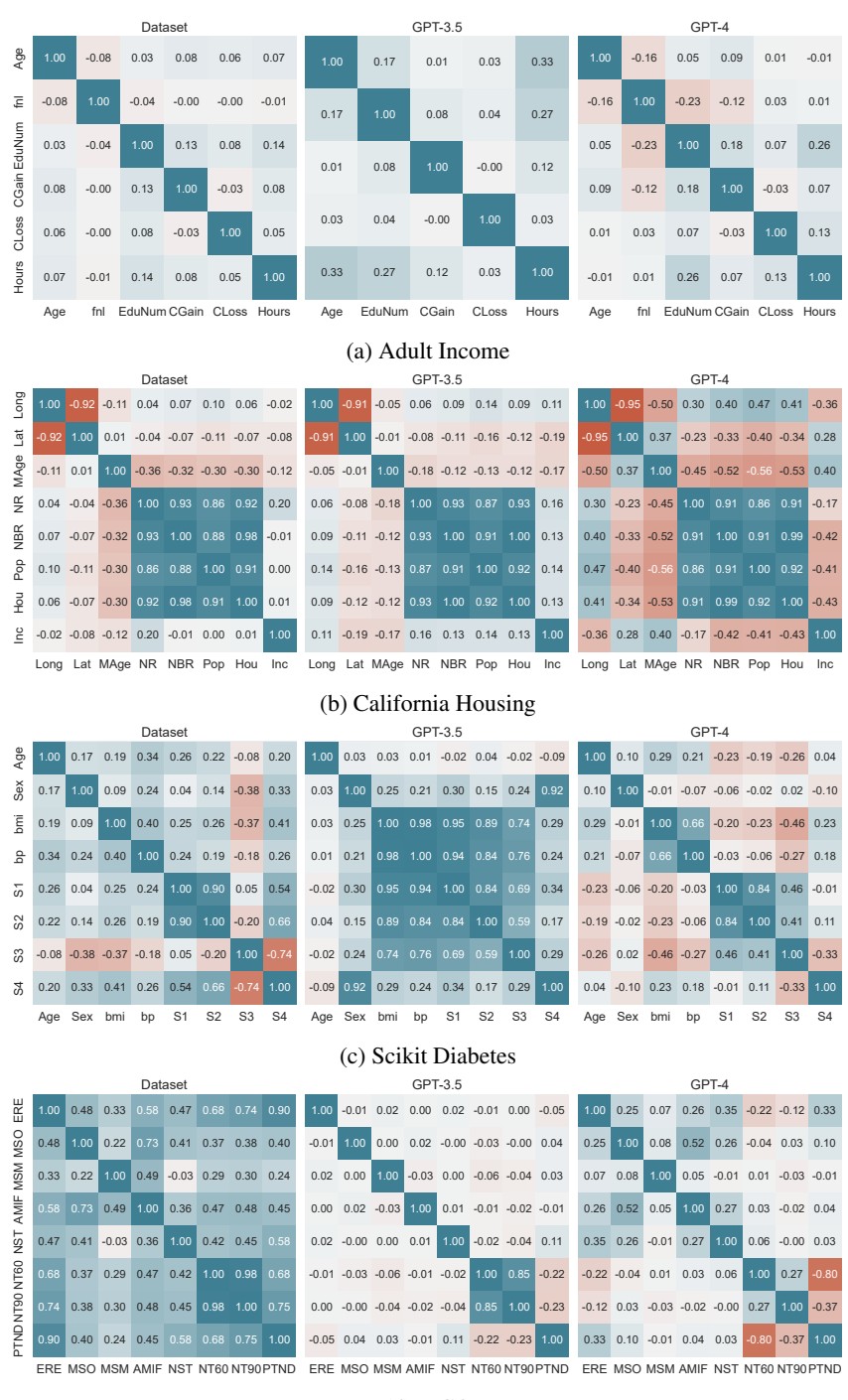

Figure 5: Pearson Correlation coefficients for all possible combinations of dataset and language model. Compare Figure 3 in the main paper. Temperature 0.7.

# E  PROMPTS

---

**System:** You are an expert assistant for tabular datasets. Your task is to list the names of the features of different datasets. The user provides a description of the dataset and some of the feature names. You then provide the names of the remaining features.

**User:** Dataset: IRIS. Feature Names: sepal_length
**Assistant:** sepal_width, petal_length, petal_width, species

**User:** Dataset: adult. Feature Names: Age
**Assistant:** WorkClass, fnlwgt, Education, EducationNum, MaritalStatus, Occupation, Relationship, Race, Gender, CapitalGain, CapitalLoss, HoursPerWeek, NativeCountry, Income

**User:** Dataset: titanic-train. Feature Names: PassengerId
**Assistant:** Survived, Pclass, Name, Sex, Age, SibSp, Parch, Ticket, Fare, Cabin, Embarked

**User:** Dataset: uci-wine. Feature Names: target
**Assistant:** alcohol, malic_acid, ash, alcalinity_of_ash, magnesium, total_phenols, flavanoids, nonflavanoid_phenols, proanthocyanins, color_intensity, hue, od280_od315_of_diluted_wines, proline

**User:** Dataset: california-housing. Feature Names: longitude
**Assistant:** latitude, housing_median_age, total_rooms, total_bedrooms, population, households, median_income, median_house_value, ocean_proximity

**User:** Dataset: fico. Feature Names: RiskPerformance

**Response:** ExternalRiskEstimate, MSinceOldestTradeOpen, MSinceMostRecentTradeOpen, AverageMInFile, NumSatisfactoryTrades, NumTrades60Ever2DerogPubRec, NumTrades90Ever2DerogPubRec, PercentTradesNeverDelq, MSinceMostRecentDelq, MaxDelq2PublicRecLast12M, MaxDelqEver, NumTotalTrades, NumTradesOpeninLast12M, PercentInstallTrades, MSinceMostRecentInqexcl7days, NumInqLast6M, NumInqLast6Mexcl7days, NetFractionRevolvingBurden, NetFractionInstallBurden, NumRevolvingTradesWBalance, NumInstallTradesWBalance, NumBank2NatlTradesWHighUtilization, PercentTradesWBalance

---

Figure 6: Feature Names.

**System:** You are a helpful autocomplete bot for tabular datasets. You will be prompted with a portion of a sample from a machine learning dataset. Your task is to complete the observation from the dataset. The observations are presented to you in the format "Feature Name = Feature Value".

**User:** Dataset: IRIS
Feature Names: sepal_length, sepal_width, petal_length, petal_width, species
Feature Values: sepal_length = 6.7, sepal_width = 3.1
**Assistant:** petal_length = 5.6, petal_width = 2.4, species = Iris-virginica

**User:** Dataset: titanic-train
Feature Names: PassengerId, Survived, Pclass, Name, Sex, Age, SibSp, Parch, Ticket, Fare, Cabin, Embarked
Feature Values: PassengerId = 542, Survived = 0, Pclass = 3, Name = Andersson, Miss. Ingeborg Constanzia
**Assistant:** Sex = female, Age = 9.0, SibSp = 4, Parch = 2, Ticket = 347082, Fare = 31.275, Cabin = nan, Embarked = S

[..]

**User:** Dataset: california-housing
Feature Names: longitude, latitude, housing_median_age, total_rooms, total_bedrooms, population, households, median_income, median_house_value, ocean_proximity
Feature Values: longitude = -118.03, latitude = 33.87, housing_median_age = 16.0, total_rooms = 2306.0
**Assistant:** total_bedrooms = 393.0, population = 1368.0, households = 387.0, median_income = 5.93, median_house_value = 277600.0, ocean_proximity = <1H OCEAN

**User:** Dataset: adult
Feature Names: Age, WorkClass, fnlwgt, Education, EducationNum, MaritalStatus, Occupation, Relationship, Race, Gender, CapitalGain, CapitalLoss, HoursPerWeek, NativeCountry, Income
Feature Values: Age = 51, WorkClass = Private, fnlwgt = 243361, Education = Some-college, EducationNum = 10, MaritalStatus = Widowed
**Response:** Occupation = Adm-clerical, [...]

Figure 7: Conditional Completion. See Figure 1 in the main paper.

**System:** You are an expert assistant for tabular datasets. You provide random samples from different datasets. The user provides the name of the dataset and the names of the features. You then provide a sample from the dataset.

**User:** Dataset: IRIS
Feature Names: sepal_length, sepal_width, petal_length, petal_width, species
**Assistant:** sepal_length = 5.1, sepal_width = 3.5, petal_length = 1.4, petal_width = 0.3, species = Iris-setosa

**User:** Dataset: adult
Feature Names: Age, WorkClass, fnlwgt, Education, EducationNum, MaritalStatus, Occupation, Relationship, Race, Gender, CapitalGain, CapitalLoss, HoursPerWeek, NativeCountry, Income
Feature Names: Age, WorkClass, fnlwgt, Education, EducationNum, MaritalStatus, Occupation, Relationship, Race, Gender, CapitalGain, CapitalLoss, HoursPerWeek, NativeCountry, Income
**Assistant:** Age = 43, WorkClass = Self-emp-inc, fnlwgt = 196945, Education = HS-grad, EducationNum = 9, MaritalStatus = Married-civ-spouse, Occupation = Other-service, Relationship = Husband, Race = Asian-Pac-Islander, Gender = Male, CapitalGain = 0, CapitalLoss = 0, HoursPerWeek = 78, NativeCountry = Thailand, Income = $<= 50K$

**User:** Dataset: openml-diabetes
Feature Names: Pregnancies, Glucose, BloodPressure, SkinThickness, Insulin, BMI, DiabetesPedigreeFunction, Age, Outcome
**Assistant:** Pregnancies = 1, Glucose = 95, BloodPressure = 74, SkinThickness = 21, Insulin = 73, BMI = 25.9, DiabetesPedigreeFunction = 0.673, Age = 36, Outcome = 0

**User:** Dataset: uci-wine
Feature Names: target, alcohol, malic_acid, ash, alcalinity_of_ash, magnesium, total_phenols, flavanoids, nonflavanoid_phenols, proanthocyanins, color_intensity, hue, od280_od315_of_diluted_wines, proline
**Assistant:** target = 1, alcohol = 13.24, malic_acid = 2.59, ash = 2.87, alcalinity_of_ash = 21.0, magnesium = 118, total_phenols = 2.8, flavanoids = 2.69, nonflavanoid_phenols = 0.39, proanthocyanins = 1.82, color_intensity = 4.32, hue = 1.04, od280_od315_of_diluted_wines = 2.93, proline = 735

**User:** Dataset: california-housing
Feature Names: longitude, latitude, housing_median_age, total_rooms, total_bedrooms, population, households, median_income, median_house_value, ocean_proximity
**Assistant:** longitude = -122.12, latitude = 37.68, housing_median_age = 45.0, total_rooms = 2179.0, total_bedrooms = 401.0, population = 1159.0, households = 399.0, median_income = 3.4839, median_house_value = 180600.0, ocean_proximity = NEAR BAY

**User:** Dataset: titanic-train
Feature Names: PassengerId, Survived, Pclass, Name, Sex, Age, SibSp, Parch, Ticket, Fare, Cabin, Embarked

**Response:** PassengerId = 746, Survived = 0, Pclass = 3, Name = Abbott, Mr. Rossmore Edward, Sex = male, Age = 16.0, SibSp = 1, Parch = 1, Ticket = C.A. 2673, Fare = 20.25, Cabin = NaN, Embarked = S

Figure 8: Zero-Knowledge Sampling.

**System:** You are an autocomplete bot for tabular datasets. You will be prompted with parts of a tabular dataset. Your task is to complete the dataset.

**User:** sepal_length,sepal_width,petal_length,petal_width,species
5.1,3.5,1.4,0.2,Iris-setosa
4.9,3,1.4,0.2,Iris-setosa
4.7,3.2,1.3,0.2,Iris-setosa
[...]
4.8,3,1.4,0.1,Iris-setosa
4.3,3,1.1,0.1,Iris-setosa
5.8,4,1.2,0.2,Iris-s
**Assistant:** etosa
5.7,4.4,1.5,0.4,Iris-setosa
5.4,3.9,1.3,0.4,Iris-setosa
5.1,3.5,1.4,0.3,Iris-setosa
[...]

**User:** longitude,latitude,housing_median_age,[...],ocean_proximity
-122.23,37.88,41.0,880.0,129.0,322.0,126.0,8.3252,452600.0,NEAR BAY
-122.22,37.86,21.0,7099.0,1106.0,2401.0,1138.0,8.3014,358500.0,NEAR BAY
-122.24,37.85,52.0,1467.0,190.0,496.0,177.0,7.2574,352100.0,NEAR BAY
-122.25,37.85,52.0,1274.0,235.0,558.0,219.0,5.6431,341300.0,NEAR BAY
-122.25,37.85,52.0,1627.0,280.0,565.0,259.0,3.
**Assistant:** 8462,342200.0,NEAR BAY
-122.25,37.85,52.0,919.0,213.0,413.0,193.0,4.0368,269700.0,NEAR BAY
-122.25,37.84,52.0,2535.0,489.0,1094.0,514.0,3.6591,299200.0,NEAR BAY
-122.25,37.84,52.0,3104.0,687.0,1157.0,647.0,3.12,241400.0,NEAR BAY
-122.26,37.84,42.0,2555.0,665.0,1206.0,595.0,2.0804,226700.0,NEAR BAY
-122.25,37.84,52.0,3549.0,707.0,1551.0,714.0,3.6912,261100.0,NEAR BAY
-122.26,37.85,52.0,2202.0,434.0,910.0,402.0,3.2031,281500.0,NEAR BAY
-122.26,37.85,52.0,3503.0,752.0,1504.0,734.0,3.2705,241800.0,N

**User:** Age,WorkClass,fnlwgt,Education,[...],Income
39, State-gov,77516, Bachelors,13, Never-married, [...],0,40, United-States, <=50K
50, Self-emp-not-inc,83311, Bachelors,13, Married-civ-spouse, [...],0,13, United-States, <=50K
38, Private,215646, HS-grad,9, Divorced, Handlers-cleane

**Response:** rs, Not-in-family, White, Male,0,0,40, United-States, <=50K
53, Private,234721, 11th,7, Married-civ-spouse, [...],0,40, United-States, <=50K
28, Private,338409, Bachelors,13, Married-civ-spouse, [...],0,40, Cuba, <=50K
37, Private,284582, Masters,14, Married-civ-spouse, [...],0,40, United-States, <=50K
49, Private,160187, 9th,5, Married-spouse-absent, [...],0,16, Jamaica, <=50K
52, Self-emp-not-inc,209642, HS-grad,9, Married-civ-spouse, [...],0,45, United-States, >50K
[...]

Figure 9: Header Test.

**System:** You are a helpful autocomplete bot for tabular datasets. Your task is to provide rows as they are contained in tabular datasets. The user provides a number of contiguous rows from a tabular dataset. You then provide the next row from the dataset.

**User:** 267,0,3,"Panula, Mr. Ernesti Arvid",male,16,4,1,3101295,39.6875,,S
268,1,3,"Persson, Mr. Ernst Ulrik",male,25,1,0,347083,7.775,,S
269,1,1,"Graham, Mrs. William Thompson (Edith Junkins)",female,58,0,1,PC 17582,153.4625,C125,S
[...]
279,0,3,"Rice, Master. Eric",male,7,4,1,382652,29.125,,Q
280,1,3,"Abbott, Mrs. Stanton (Rosa Hunt)",female,35,1,1,C.A. 2673,20.25,,S
281,0,3,"Duane, Mr. Frank",male,65,0,0,336439,7.75,,Q
**Assistant:** 282,0,3,"Olsson, Mr. Nils Johan Goransson",male,28,0,0,347464,7.8542,,S

[...]

**User:** 142,1,3,"Nysten, Miss. Anna Sofia",female,22,0,0,347081,7.75,,S
143,1,3,"Hakkarainen, Mrs. Pekka Pietari (Elin Matilda Dolck)",female,24,1,0,STON/O2. 3101279,15.85,,S
144,0,3,"Burke, Mr. Jeremiah",male,19,0,0,365222,6.75,,Q
145,0,2,"Andrew, Mr. Edgardo Samuel",male,18,0,0,231945,11.5,,S
[...]
154,0,3,"van Billiard, Mr. Austin Blyler",male,40.5,0,2,A/5. 851,14.5,,S
155,0,3,"Olsen, Mr. Ole Martin",male,,0,0,Fa 265302,7.3125,,S
156,0,1,"Williams, Mr. Charles Duane",male,51,0,1,PC 17597,61.3792,,C
**Assistant:** 157,1,3,"Gilnagh, Miss. Katherine ""Katie""",female,16,0,0,35851,7.7333,,Q

**User:** 613,1,3,"Murphy, Miss. Margaret Jane",female,,1,0,367230,15.5,,Q
614,0,3,"Horgan, Mr. John",male,,0,0,370377,7.75,,Q
615,0,3,"Brocklebank, Mr. William Alfred",male,35,0,0,364512,8.05,,S
616,1,2,"Herman, Miss. Alice",female,24,1,2,220845,65,,S
617,0,3,"Danbom, Mr. Ernst Gilbert",male,34,1,1,347080,14.4,,S
618,0,3,"Lobb, Mrs. William Arthur (Cordelia K Stanlick)",female,26,1,0,A/5. 3336,16.1,,S
619,1,2,"Becker, Miss. Marion Louise",female,4,2,1,230136,39,F4,S
620,0,2,"Gavey, Mr. Lawrence",male,26,0,0,31028,10.5,,S
621,0,3,"Yasbeck, Mr. Antoni",male,27,1,0,2659,14.4542,,C
622,1,1,"Kimball, Mr. Edwin Nelson Jr",male,42,1,0,11753,52.5542,D19,S
623,1,3,"Nakid, Mr. Sahid",male,20,1,1,2653,15.7417,,C
624,0,3,"Hansen, Mr. Henry Damsgaard",male,21,0,0,350029,7.8542,,S
625,0,3,"Bowen, Mr. David John ""Dai""",male,21,0,0,54636,16.1,,S
626,0,1,"Sutton, Mr. Frederick",male,61,0,0,36963,32.3208,D50,S
627,0,2,"Kirkland, Rev. Charles Leonard",male,57,0,0,219533,12.35,,Q

**Response:** 628,0,3,"Bostandyeff, Mr. Guentcho",male,26,0,0,349224,7.8958,,S

Figure 10: Row Completion and First Token Test.

**System:** You are a helpful autocomplete bot for tabular datasets. You will be prompted with a portion of a sample from a machine learning dataset. Your task is to complete the observation from the dataset. The observations are presented to you in the format "Feature Name = Feature Value".

**User:** Age = 22, WorkClass = ?, Education = Some-college, EducationNum = 10, MaritalStatus = Divorced, Occupation = ?, Relationship = Not-in-family, Race = White, Gender = Female, CapitalGain = 0, CapitalLoss = 0, HoursPerWeek = 35, NativeCountry = United-States, Income = <= 50K

**Assistant:** fnlwgt = 376277

**User:** Age = 78, WorkClass = Self-emp-inc, Education = Bachelors, EducationNum = 13, MaritalStatus = Married-civ-spouse, Occupation = Exec-managerial, Relationship = Husband, Race = White, Gender = Male, CapitalGain = 0, CapitalLoss = 2392, HoursPerWeek = 40, NativeCountry = United-States, Income = > 50K

**Assistant:** fnlwgt = 188044

**User:** Age = 32, WorkClass = Federal-gov, Education = Some-college, EducationNum = 10, MaritalStatus = Divorced, Occupation = Adm-clerical, Relationship = Unmarried, Race = White, Gender = Female, CapitalGain = 0, CapitalLoss = 0, HoursPerWeek = 40, NativeCountry = United-States, Income = <= 50K

**Assistant:** fnlwgt = 566117

**User:** Age = 35, WorkClass = Private, Education = Bachelors, EducationNum = 13, MaritalStatus = Never-married, Occupation = Prof-specialty, Relationship = Not-in-family, Race = White, Gender = Male, CapitalGain = 0, CapitalLoss = 0, HoursPerWeek = 50, NativeCountry = United-States, Income = > 50K

**Assistant:** fnlwgt = 377798

**User:** Age = 32, WorkClass = Private, Education = Some-college, EducationNum = 10, MaritalStatus = Divorced, Occupation = Other-service, Relationship = Other-relative, Race = White, Gender = Male, CapitalGain = 0, CapitalLoss = 0, HoursPerWeek = 40, NativeCountry = Nicaragua, Income = <= 50K

**Assistant:** fnlwgt = 160362

**User:** Age = 28, WorkClass = Federal-gov, Education = HS-grad, EducationNum = 9, MaritalStatus = Never-married, Occupation = Craft-repair, Relationship = Not-in-family, Race = White, Gender = Male, CapitalGain = 0, CapitalLoss = 0, HoursPerWeek = 40, NativeCountry = United-States, Income = <= 50K

**Response:** fnlwgt = 183175

Figure 11: Feature Completion Test.

**System:** You are a helpful statistician, data scientist, and board-certified medical doctor with expertise in treating diabetes.

You help to make predictions on the openml diabetes dataset. This dataset contains the following features: Pregnancies, Glucose, BloodPressure, SkinThickness, Insulin, BMI, DiabetesPedigree-Function, and Age.

Your task is to predict whether a patient tested positive or negative for diabetes.

The user provides you with the data of different individuals. You respond with '1' if your assessment is that the patient tested positive for diabetes, and with '0' if your assessment is that the patient tested negative for diabetes.

Read all the provided inputs carefully and provide your best overall assessment.

**User:** IF Pregnancies = 5.0, Glucose = 136.0, BloodPressure = 82.0, SkinThickness = 0.0, Insulin = 0.0, BMI = 0.0, DiabetesPedigreeFunction = 0.64, Age = 69.0, THEN
**Assistant:** 0

[...]

**User:** IF Pregnancies = 5.0, Glucose = 115.0, BloodPressure = 76.0, SkinThickness = 0.0, Insulin = 0.0, BMI = 31.2, DiabetesPedigreeFunction = 0.343, Age = 44.0, THEN
**Assistant:** 1

**User:** IF Pregnancies = 6.0, Glucose = 148.0, BloodPressure = 72.0, SkinThickness = 35.0, Insulin = 0.0, BMI = 33.6, DiabetesPedigreeFunction = 0.627, Age = 50.0, THEN

**Response:** 1

Figure 12: Prompt structure for the prediction task in Table 2 in the main paper.

## F    ADDITIONAL FIGURES FOR THE REBUTTAL

```
Dataset: iris
Model: togethercomputer/llama-2-7b
Few-Shot: adult-train, titanic-train, uci-wine, california-housing, openml-diabetes
Feature Names Test
Feature Names:    sepal_width, petal_length, petal_width, species
Model Generation: sepal_width, petal_length, petal_width, species
Header Test: sepal_length,sepal_width,petal_length,petal_width,species
5.1,3.5,1.4,0.2,Iris-setosa
4.9,3,1.4,0.2,Iris-setosa
4.7,3.2,1.3,0.2,Iris-setosa
4.6,3.1,1.5,0.2,Iris-setosa
5,3.6,1.4,0.2,Iris-setosa
5.1,3.5,1.4,0.2,Iris-setosa
4.9,3,1.4,0.2,Iris-setosa
4.7,3.2,1.3,0.2,Iris-setosa
4.6,3.1,1.5,0.2,Iris-setosa
5,3.6,1.4,0.2,Iris-setosa
5.1,3.5,1.4,0.2,Iris-setosa
4.9,3,1.4,0.2,Iris-setosa
4.7,3.2,1.3,0.2,Iris-setosa
4.6,3.1,1.5,0.2,Iris-setosa
5,3.6,1.4,0.2,Iris-setosa
5.1,3.5,1.4,0.2,Iris-setosa
4.9,3,1.4,0.2,Iris-setosa
4.7,3.2,1.3,0.2,Iris-setosa
4.6,3.1,1.5,0.2,Iris-setosa
5,3.6,1.4,0.2,Iris-setosa
5.1,3.5,1.4,0.2,Iris-setosa
4.9,3,1.4,0.2,Iris-setosa
4.7,3.2,1.3,0.2,Iris-setosa
4.6,3.1,1.5,0.2,Iris-setosa
5,3.6,1.4,0.2,Iris-setosa
5.1,3.5,1.4,0.2,Iris-setosa
4.9,3,1.4,0.2,Iris-setosa
4.7,3.2,1.3,0.2,Iris
Header Test Legend:  Prompt Correct Incorrect
Drawing 10 zero-knowledge samples at temperature 0.7:
  sepal_length sepal_width petal_length petal_width species
5         1.4          0.2          1.2          0.2        0
8         5.1          3.5          1.4          0.2        3
The model provided 2 valid samples.
Info: 1.99% of the rows in this dataset are duplicates.
Row Completion Test: 0/25 exact matches.
Info: Using feature petal_length with 28.67% unique values.
Feature Completion Test ("petal_length"): 2/25 exact matches.
First Token Test: 9/25 exact matches.
First Token Test Baseline (Matches of most common first token): 12/25.
```

Figure 13: Detailed test results of Llama-2-7b on Iris. The model knows the feature names and the format of the feature values. It can complete a single row in the header test. Because the header test measures the best of 5 completions, this hardly counts as evidence of memorization. The model fails the row completion, feature completion, and first token test. The results in this Figure should be compared to the results with more powerful LLMs in the subsequent figures.

```
Dataset: iris
Model: togethercomputer/llama-2-70b
Few-Shot: adult-train, titanic-train, uci-wine, california-housing, openml-diabetes
Feature Names Test
Feature Names:    sepal_width, petal_length, petal_width, species
Model Generation: sepal_width, petal_length, petal_width, species
Header Test: sepal_length,sepal_width,petal_length,petal_width,species
5.1,3.5,1.4,0.2,Iris-setosa
4.9,3,1.4,0.2,Iris-setosa
4.7,3.2,1.3,0.2,Iris-setosa
4.6,3.1,1.5,0.2,Iris-setosa
5,3.6,1.4,0.2,Iris-setosa
5.4,3.9,1.7,0.4,Iris-setosa
4.6,3.4,1.4,0.3,Iris-setosa
5,3.4,1.5,0.2,Iris-setosa
4.4,2.9,1.4,0.2,Iris-setosa
4.9,3.1,1.5,0.1,Iris-setosa
5.4,3.7,1.5,0.2,Iris-setosa
4.8,3.4,1.6,0.2,Iris-setosa
4.8,3,1.4,0.1,Iris-setosa
4.3,3,1.1,0.1,Iris-setosa
5.8,4,1.2,0.2,Iris-setosa
5.7,4.4,1.5,0.4,Iris-setosa
5.4,3.9,1.3,0.4,Iris-setosa
5.1,3.5,1.4,0.3,Iris-setosa
5.7,3.8,1.7,0.3,Iris-setosa
5.1,3.8,1.5,0.3,Iris-setosa
5.4,3.4,1.7,0.2,Iris-setosa
5.1,3.7,1.5,0.4,Iris-setosa
4.6,3.6,1,0.2,Iris-setosa
5.1,3.3,1.7,0.5,Iris-setosa
4.8,3,1.4,0.1,Iris-setosa
4.5,2.3,1.3,0.3,Iris-setosa
4.4,3,1.3,0.2,Iris-setosa
4.7,3.2,1.6,0.2,Iris
Header Test Legend:  Prompt Correct Incorrect
Drawing 10 zero-knowledge samples at temperature 0.7:
  sepal_length sepal_width petal_length petal_width species
0          5.2          3.5          1.5         0.2  setosa
2          5.1          3.3          1.7         0.5        0
3          5.1          3.5          1.4         0.2  setosa
5          4.9          2.4          3.3         1.1  setosa
6          5.1          3.5          1.4         0.2  setosa
8          5.1          3.5          1.4         0.2  setosa
The model provided 6 valid samples.
Info: 1.99% of the rows in this dataset are duplicates.
Row Completion Test: 8/25 exact matches.
Info: Using feature petal_length with 28.67% unique values.
Feature Completion Test ("petal_length"): 2/25 exact matches.
First Token Test: 19/25 exact matches.
First Token Test Baseline (Matches of most common first token): 14/25.
```

Figure 14: Detailed test results of Llama-2-70b on Iris. The model knows the feature names and the format of the feature values. The header test provides clear evidence of memorization. The row completion and first token test also provide evidence for memorization. However, the results of these two tests are marginal and hard to interpret because of the limited amount of entropy in the Iris dataset.

```
Dataset: iris
Model: gpt-3.5-turbo-16k-0613
Few-Shot: adult-train, titanic-train, uci-wine, california-housing, openml-diabetes
Feature Names Test
Feature Names:    sepal_width, petal_length, petal_width, species
Model Generation: sepal_width, petal_length, petal_width, species
Header Test: sepal_length,sepal_width,petal_length,petal_width,species
5.1,3.5,1.4,0.2,Iris-setosa
4.9,3,1.4,0.2,Iris-setosa
4.7,3.2,1.3,0.2,Iris-setosa
4.6,3.1,1.5,0.2,Iris-setosa
5,3.6,1.4,0.2,Iris-setosa
5.4,3.9,1.7,0.4,Iris-setosa
4.6,3.4,1.4,0.3,Iris-setosa
5,3.4,1.5,0.2,Iris-setosa
4.4,2.9,1.4,0.2,Iris-setosa
4.9,3.1,1.5,0.1,Iris-setosa
5.4,3.7,1.5,0.2,Iris-setosa
4.8,3.4,1.6,0.2,Iris-setosa
4.8,3,1.4,0.1,Iris-setosa
4.3,3,1.1,0.1,Iris-setosa
5.8,4,1.2,0.2,Iris-setosa
5.7,4.4,1.5,0.4,Iris-setosa
5.4,3.9,1.3,0.4,Iris-setosa
5.1,3.5,1.4,0.3,Iris-setosa
5.7,3.8,1.7,0.3,Iris-setosa
5.1,3.8,1.5,0.3,Iris-setosa
5.4,3.4,1.7,0.2,Iris-setosa
5.1,3.7,1.5,0.4,Iris-setosa
4.6,3.6,1,0.2,Iris-setosa
5.1,3.3,1.7,0.5,Iris-setosa
4.8,3.4,1.9,0.2,Iris-setosa
5,3,1.6,0.2,Iris-setosa
5,3.4,1.6,0.4,Iris-setosa
5.2,3.5,1.5,0.2,Iris-setosa
5.2,3.4,1.4
Header Test Legend:  Prompt Correct Incorrect
Drawing 10 zero-knowledge samples at temperature 0.7:
  sepal_length sepal_width petal_length petal_width     species
0          5.1         3.5          1.4         0.2      setosa
1          5.1         3.5          1.4         0.2      setosa
2          6.4         3.2          4.5         1.5  versicolor
3          5.1         3.5          1.4         0.2      setosa
4          5.1         3.5          1.4         0.2      setosa
5          5.1         3.5          1.4         0.2      setosa
6          5.1         3.5          1.4         0.2      setosa
7          5.1         3.5          1.4         0.2      setosa
8          5.1         3.5          1.4         0.2      setosa
9          5.1         3.5          1.4         0.2      setosa
Info: 1.99% of the rows in this dataset are duplicates.
Row Completion Test: 8/25 exact matches.
Info: Using feature petal_length with 28.67% unique values.
Feature Completion Test ("petal_length"): 19/25 exact matches.
First Token Test: 8/25 exact matches.
First Token Test Baseline (Matches of most common first token): 9/25.
```

Figure 15: Detailed test results of gpt-3.5-turbo-16k-0613 on Iris. While there are differences, the results are overall comparable to the results with Llama-2-70b depicted on the previous page.

```
Dataset: iris
Model: gpt-4-0613
Few-Shot: adult-train, titanic-train, uci-wine, california-housing, openml-diabetes
Feature Names Test
Feature Names:    sepal_width, petal_length, petal_width, species
Model Generation: sepal_width, petal_length, petal_width, species
Header Test: sepal_length,sepal_width,petal_length,petal_width,species
5.1,3.5,1.4,0.2,Iris-setosa
4.9,3,1.4,0.2,Iris-setosa
4.7,3.2,1.3,0.2,Iris-setosa
4.6,3.1,1.5,0.2,Iris-setosa
5,3.6,1.4,0.2,Iris-setosa
5.4,3.9,1.7,0.4,Iris-setosa
4.6,3.4,1.4,0.3,Iris-setosa
5,3.4,1.5,0.2,Iris-setosa
4.4,2.9,1.4,0.2,Iris-setosa
4.9,3.1,1.5,0.1,Iris-setosa
5.4,3.7,1.5,0.2,Iris-setosa
4.8,3.4,1.6,0.2,Iris-setosa
4.8,3,1.4,0.1,Iris-setosa
4.3,3,1.1,0.1,Iris-setosa
5.8,4,1.2,0.2,Iris-setosa
5.7,4.4,1.5,0.4,Iris-setosa
5.4,3.9,1.3,0.4,Iris-setosa
5.1,3.5,1.4,0.3,Iris-setosa
5.7,3.8,1.7,0.3,Iris-setosa
5.1,3.8,1.5,0.3,Iris-setosa
5.4,3.4,1.7,0.2,Iris-setosa
5.1,3.7,1.5,0.4,Iris-setosa
4.6,3.6,1,0.2,Iris-setosa
5.1,3.3,1.7,0.5,Iris-setosa
4.8,3.4,1.9,0.2,Iris-setosa
5,3,1.6,0.2,Iris-setosa
5,3.4,1.6,0.4,Iris-setosa
5.2,3.5,1.5,0.2,Iris-setosa
5.2,
Header Test Legend:  Prompt Correct Incorrect
Drawing 10 zero-knowledge samples at temperature 0.7:
  sepal_length sepal_width petal_length petal_width species
0          5.1          3.5          1.4          0.2  setosa
1          5.1          3.5          1.4          0.2  setosa
2          5.1          3.5          1.4          0.2  setosa
3          5.1          3.5          1.4          0.2  setosa
4          5.1          3.5          1.4          0.2  setosa
5          5.1          3.5          1.4          0.2  setosa
6          5.1          3.5          1.4          0.2  setosa
7          5.1          3.5          1.4          0.2  setosa
8          5.1          3.5          1.4          0.2  setosa
9          5.1          3.5          1.4          0.2  setosa
Info: 1.99% of the rows in this dataset are duplicates.
Row Completion Test: 24/25 exact matches.
Info: Using feature petal_length with 28.67% unique values.
Feature Completion Test ("petal_length"): 20/25 exact matches.
First Token Test: 25/25 exact matches.
First Token Test Baseline (Matches of most common first token): 13/25.
```

Figure 16: Detailed test results of gpt-4-0613 on Iris. The model shows unambiguous evidence of memorization in all the different tests.

```
Dataset: titanic-train
Model: gpt-3.5-turbo-1106
Few-Shot: iris, adult-train, openml-diabetes, uci-wine, california-housing
Feature Names Test
Feature Names:    Name, Sex, Age, SibSp, Parch, Ticket, Fare, Cabin, Embarked
Model Generation: Name, Sex, Age, SibSp, Parch, Ticket, Fare, Cabin, Embarked
Header Test: PassengerId,Survived,Pclass,Name,Sex,Age,SibSp,Parch,Ticket,Fare,Cabin,Embarked
1,0,3,"Braund, Mr. Owen Harris",male,22,1,0,A/5 21171,7.25,,S
2,1,1,"Cumings, Mrs. John Bradley (Florence Briggs Thayer)",female,38,1,0,PC 17599,71.2833,C85,C
3,1,3,"Heikkinen, Miss. Laina",female,26,0,0,STON/O2. 3101282,7.925,,S
4,1,1,"Futrelle, Mrs. Jacques Heath (Lily May Peel)",female,35,1,0,113803,53.1,C123,S
5,0,3,"Allen, Mr. William Henry",male,35,0,0,373450,8.05,,S
6,0,3,"Moran, Mr. James",male,,0,0,330877,8.4583,,Q
7,0,1,"McCarthy, Mr. Timothy J",male,54,0,0,17463,51.8625,E46,S
8,0,3,"Palsson, Master. Gosta Leonard",male,2,3,1,349909,21.075,,S
Header Test Legend:  Prompt Correct Incorrect
Drawing 10 zero-knowledge samples at temperature 0.7:
   PassengerId Survived Pclass                                  Name     Sex   Age SibSp Parch   Ticket      Fare Cabin Embarked
0          493        0      1          Molson, Mr. Harry Markland      male  55.0     0     0   113787      30.5   C30        S
1          556        1      1                 "Wright, Mr. George"     male    62     0     0   113807     26.55   NaN        S
2          521        1      1               Perreault, Miss. Anne   female  30.0     0     0    12749      93.5   B73        S
3          342        0      2               Collander, Mr. Erik Gustaf  male  28.0     0     0   248740      13.0    F2        S
4          876        1      3  "Najib, Miss. Adele Kiamie ""Jane"""  female    15     0     0     2667     7.225  None        C
5          356        0      3               Vanden Steen, Mr. Leo Peter  male  28.0     0     0   345783       9.5  None        S
6            7        0      1                 McCarthy, Mr. Timothy J   male    54     0     0    17463   51.8625   E46        S
7          543        1      2      "Beane, Mrs. Edward (Ethel Clarke)"  female    19     1     0     2908      26.0  E101        S
8          731        1      1           Allen, Miss. Elisabeth Walton  female  29.0     0     0    24160  211.3375    B5        S
9          543        1      2                    "Beane, Mr. Edward"     male    32     1     0     2908      26.0  null        S
Info: All the rows in the dataset are unique.
Row Completion Test: 20/25 exact matches.
Feature Completion Test ("Name"): 24/25 exact matches.
Info: Aborting the first token test because the first token does not seem to be random.
The most likely reason for this is that the rows in the csv file are not random.
For example, the first feature might be the id of the observation.
```

Figure 17: Detailed test results of gpt-3.5-turbo-1106 on the train split of the Kaggle Titanic dataset. As for the results reported in the main paper, there is unambiguous evidence of memorization (note that this dataset contains a high amount of entropy, especially in the unique Name feature). Best viewed in digital format.

```
Dataset: titanic-test
Model: gpt-3.5-turbo-1106
Few-Shot: iris, adult-train, openml-diabetes, uci-wine, california-housing
Feature Names Test
Feature Names:    Name, Sex, Age, SibSp, Parch, Ticket, Fare, Cabin, Embarked
Model Generation: Name, Sex, Age, SibSp, Parch, Ticket, Fare, Cabin, Embarked
Header Test: PassengerId,Pclass,Name,Sex,Age,SibSp,Parch,Ticket,Fare,Cabin,Embarked
892,3,"Kelly, Mr. James",male,34.5,0,0,330911,7.8292,,Q
893,3,"Wilkes, Mrs. James (Ellen Needs)",female,47,1,0,363272,7,,S
894,2,"Myles, Mr. Thomas Francis",male,62,0,0,240276,9.6875,,Q
895,3,"Wirz, Mr. Albert",male,27,0,0,315154,8.6625,,S
896,3,"Hirvonen, Mrs. Alexander (Helga E Lindqvist)",female,22,1,1,3101298,12.2875,,S
897,3,"Svensson, Mr. Johan Cervin",male,14,0,0,7538,9.225,,S
898,3,"Connolly, Miss. Kate",female,30,0,0,330972,7.6292,,Q
899,2,"Caldwell, Mr. Albert Francis",male,26,1,1,248738,29,,S
Legend:  Prompt Correct Incorrect
Drawing 10 zero-knowledge samples at temperature 0.7:
   PassengerId Pclass                                 Name     Sex   Age SibSp Parch      Ticket     Fare Cabin Embarked
0          892      3                    "Kelly, Mr. James"     male  34.5     0     0      330911   7.8292                Q
1          892      3                    "Kelly, Mr. James"     male  34.5     0     0      330911   7.8292                Q
2          892      3                    "Kelly, Mr. James"     male  34.5     0     0      330911   7.8292   null          Q
3          892      3                    "Kelly, Mr. James"     male  34.5     0     0      330911   7.8292   None          Q
4          892      3                    "Kelly, Mr. James"     male  34.5     0     0      330911   7.8292                Q
5          892      3                    "Kelly, Mr. James"     male  34.5     0     0      330911   7.8292   null          Q
6          892      3                    "Kelly, Mr. James"     male  34.5     0     0      330911   7.8292                Q
7          432      3  "Thorne, Mrs. Gertrude Maybelle"  female    29     1     1  "W./C. 6608"   34.375   null          S
8          892      3                    "Kelly, Mr. James"     male  34.5     0     0      330911   7.8292                Q
9          892      3                    "Kelly, Mr. James"     male  34.5     0     0      330911   7.8292   null          Q
Info: All the rows in the dataset are unique.
Row Completion Test: 14/25 exact matches.
Feature Completion Test ("Name"): 19/25 exact matches.
Info: Aborting the first token test because the first token does not seem to be random.
The most likely reason for this is that the rows in the csv file are not random.
For example, the first feature might be the id of the observation.
```

Figure 18: Detailed test results of gpt-3.5-turbo-1106 on the test split of the Kaggle Titanic dataset. As for the results reported in the main paper, there is unambiguous evidence of memorization (note that this dataset contains a high amount of entropy, especially in the unique Name feature). Best viewed in digital format.

```
Dataset: adult-train
Model: gpt-4-0613
Few-Shot: iris, titanic-train, openml-diabetes, uci-wine, california-housing
Feature Names Test
Feature Names:    Education, EducationNum, MaritalStatus, Occupation, Relationship, Race, Gender, CapitalGain, CapitalLoss, HoursPerWeek, NativeCountry, Income
Model Generation: Education, EducationNum, MaritalStatus, Occupation, Relationship, Race, Sex, CapitalGain, CapitalLoss, HoursPerWeek, NativeCountry, Income
Header Test: Age,WorkClass,fnlwgt,Education,EducationNum,MaritalStatus,Occupation,Relationship,Race,Gender,CapitalGain,CapitalLoss,HoursPerWeek,NativeCountry,Income
39, State-gov,77516, Bachelors,13, Never-married, Adm-clerical, Not-in-family, White, Male,2174,0,40, United-States, <=50K
50, Self-emp-not-inc,83311, Bachelors,13, Married-civ-spouse, Exec-managerial, Husband, White, Male,0,0,13, United-States, <=50K
38, Private,215646, HS-grad,9, Divorced, Handlers-cleaners, Not-in-family, White, Male,0,0,40, United-States, <=50K
53, Private,234721, 11th,7, Married-civ-spouse, Handlers-cleaners, Husband, Black, Male,0,0,40, United-States, <=50K
28, Private,338409, Bachelors,13, Married-civ-spouse, Prof-specialty, Wife, Black, Female,0,0,40, Cuba, <=50K
37, Private,284582, Masters,14, Married-civ-spouse, Exec-managerial, Wife, White, Female,0,0,40, United-States, <=50K
49, Private,160187, 9th,5, Married-spouse-absent, Other-service, Not-in-family, Black, Female,0,0,16, Jamaica, <=50K
52, Self-emp-not-inc,209642, HS-grad,9, Married-civ-spouse, Exec-managerial, Husband, White, Male,0,0,45, United-States, >50K
31, Private,45781, Masters,14, Never-married, Prof-specialty, Not-in-family, White, Female,14084,0,50, United-States, >50K
42, Private,159449, Bachelors,13, Married-civ-spouse, Exec-managerial, Husband, White, Male,5178,0,40, United-States, >50K
37, Private,280464, Some-college,10, Married-civ-spouse, Exec-managerial, Husband, Black, Male,0,0,80, United-States, >50K
30, State-gov,141297, Bachelors,13, Married-civ-spouse, Prof-specialty, Husband, Asian-Pac-Islander, Male,0,0,40, India, >50K
23, Private,122272, Bachelors,13, Never-married, Adm-clerical, Own-child, White, Female,0,0,30, United-States, <=50K
32, Private,205019, Assoc-acdm,12, Never-married, Sales, Not-in-family, Black, Male,0,0,50, United-States, <=50K
40, Private,121772, Assoc-voc,11, Married-civ-spouse, Craft-repair, Husband, Asian-Pac-Islander, Male,0,0,40, United-States, >50K
34, Private,245487, 7th-8th,4, Married-civ-spouse, Transport-moving, Husband, Amer-Indian-Eskimo, Male,0,
Header Test Legend:  Prompt Correct Incorrect
Drawing 10 zero-knowledge samples at temperature 0.7:
   Age  WorkClass  fnlwgt  Education EducationNum      MaritalStatus      Occupation   Relationship   Race  Gender CapitalGain CapitalLoss HoursPerWeek  NativeCountry Income
0   40    Private  215646    HS-grad           9           Divorced  Handlers-cleaners  Not-in-family  White    Male           0           0           40  United-States  <=50K
1   49    Private  101320 Assoc-acdm          12 Married-civ-spouse    Exec-managerial           Wife  White  Female           0        1902           40  United-States   >50K
2   39  State-gov   77516  Bachelors          13      Never-married       Adm-clerical  Not-in-family  White    Male        2174           0           40  United-States  <=50K
3   38    Private   89814    HS-grad           9 Married-civ-spouse       Craft-repair        Husband  White    Male           0           0           40  United-States  <=50K
4   39  State-gov   77516  Bachelors          13      Never-married       Adm-clerical  Not-in-family  White    Male        2174           0           40  United-States  <=50K
5   39  State-gov   77516  Bachelors          13      Never-married       Adm-clerical  Not-in-family  White    Male        2174           0           40  United-States  <=50K
6   39  State-gov   77516  Bachelors          13      Never-married       Adm-clerical  Not-in-family  White    Male        2174           0           40  United-States  <=50K
7   39  State-gov   77516  Bachelors          13      Never-married       Adm-clerical  Not-in-family  White    Male        2174           0           40  United-States  <=50K
8   39  State-gov   77516  Bachelors          13      Never-married       Adm-clerical  Not-in-family  White    Male        2174           0           40  United-States  <=50K
9   45    Private   98095    HS-grad           9 Married-civ-spouse       Craft-repair        Husband  White    Male           0           0           40  United-States  <=50K
Info: 0.07% of the rows in this dataset are duplicates.
Row Completion Test: 1/25 exact matches.
Info: Using feature fnlwgt with 66.48% unique values.
Feature Completion Test ("fnlwgt"): 0/25 exact matches.
First Token Test: 3/25 exact matches.
First Token Test Baseline (Matches of most common first token): 12/25.
```

Figure 19: Detailed test results of gpt-4-0613 on the train split of the Adult dataset. The results are the same as the results for the Adult dataset reported in the main paper. Best viewed in digital format.

```
Dataset: adult-test
Model: gpt-4-0613
Few-Shot: iris, titanic-train, openml-diabetes, uci-wine, california-housing
Feature Names Test
Feature Names:    Education, EducationNum, MaritalStatus, Occupation, Relationship, Race, Gender, CapitalGain, CapitalLoss, HoursPerWeek, NativeCountry, Income
Model Generation: Education, EducationNum, MaritalStatus, Occupation, Relationship, Race, Sex, CapitalGain, CapitalLoss, HoursPerWeek, NativeCountry, Income
Header Test: Age,WorkClass,fnlwgt,Education,EducationNum,MaritalStatus,Occupation,Relationship,Race,Gender,CapitalGain,CapitalLoss,HoursPerWeek,NativeCountry,Income
25, Private, 226802, 11th, 7, Never-married, Machine-op-inspct, Own-child, Black, Male, 0, 0, 40, United-States, <=50K.
38, Private, 89814, HS-grad, 9, Married-civ-spouse, Farming-fishing, Husband, White, Male, 0, 0, 50, United-States, <=50K.
28, Local-gov, 336951, Assoc-acdm, 12, Married-civ-spouse, Protective-serv, Husband, White, Male, 0, 0, 40, United-States, >50K.
44, Private, 160323, Some-college, 10, Married-civ-spouse, Machine-op-inspct, Husband, Black, Male, 7688, 0, 40, United-States, >50K.
18, ?, 103497, Some-college, 10, Never-married, ?, Own-child, White, Female, 0, 0, 30, United-States, <=50K.
34, Private, 198693, 10th, 6, Never-married, Other-service, Not-in-family, White, Male, 0, 0, 30, United-States, <=50K.
29, ?, 227026, HS-grad, 9, Never-married, ?, Unmarried, Black, Male, 0, 0, 40, United-States, <=50K.
63, Self-emp-not-inc, 104626, Prof-school, 15, Married-civ-spouse, Prof-specialty, Husband, White, Male, 3103, 0, 32, United-States, >50K.
24, Private, 369667, Some-college, 10, Never-married, Other-service, Unmarried, White, Female, 0, 0, 40, United-States, <=50K.
55, Private, 104996, 7th-8th, 4, Married-civ-spouse, Craft-repair, Husband, White, Male, 0, 0, 10, United-States, <=50K.
65, Private, 184454, HS-grad, 9, Married-civ-spouse, Machine-op-inspct, Husband, White, Male, 6418, 0, 40, United-States, >50K.
36, Federal-gov, 212465, Bachelors, 13, Married-civ-spouse, Adm-clerical, Husband, White, Male, 0, 0, 40, United-States, <=50K.
26, Private, 82091, HS-grad, 9, Never-married, Adm-clerical, Not-in-family, White, Female, 0, 0, 39, United-States, <=50K.
58, ?, 299831, HS-grad, 9, Married-civ-spouse, ?, Husband, White, Male, 0, 0, 35, United-States, <=50K.
48, Private, 279724, HS-grad, 9, Divorced, Machine-op-inspct, Unmarried, White, Male, 0, 0, 40, United-States,
Header Test Legend:  Prompt Correct Incorrect
Drawing 10 zero-knowledge samples at temperature 0.7:
   Age  WorkClass  fnlwgt  Education EducationNum      MaritalStatus      Occupation   Relationship   Race  Gender CapitalGain CapitalLoss HoursPerWeek  NativeCountry Income
0   38    Private   89814    HS-grad           9 Married-civ-spouse   Transport-moving        Husband  White    Male           0           0           40  United-States  <=50K
1   40    Private   77143  Bachelors          13 Married-civ-spouse    Exec-managerial        Husband  White    Male           0           0           40  United-States   >50K
2   38    Private  215646    HS-grad           9           Divorced  Handlers-cleaners  Not-in-family  White    Male           0           0           40  United-States  <=50K
3   38    Private  215646    HS-grad           9           Divorced  Handlers-cleaners  Not-in-family  White    Male           0           0           40  United-States  <=50K
4   45    Private   78916    Masters          14      Never-married    Exec-managerial  Not-in-family  White    Male           0           0           50  United-States   >50K
5   37    Private  215646    HS-grad           9      Never-married  Handlers-cleaners  Not-in-family  White    Male           0           0           40  United-States  <=50K
6   38    Private   89814    HS-grad           9           Divorced  Machine-op-inspct      Unmarried  White  Female           0           0           40  United-States  <=50K
7   35  State-gov   77516  Bachelors          13      Never-married       Adm-clerical  Not-in-family  White    Male        2174           0           40  United-States  <=50K
8   50    Private   83311  Bachelors          13 Married-civ-spouse    Exec-managerial        Husband  White    Male           0           0           13  United-States   >50K
9   37    Private  215646    HS-grad           9      Never-married  Handlers-cleaners  Not-in-family  White    Male           0           0           40  United-States  <=50K
Info: 0.03% of the rows in this dataset are duplicates.
Row Completion Test: 0/25 exact matches.
Info: Using feature fnlwgt with 78.54% unique values.
Feature Completion Test ("fnlwgt"): 0/25 exact matches.
First Token Test: 5/25 exact matches.
First Token Test Baseline (Matches of most common first token): 15/25.
```

Figure 20: Detailed test results of gpt-4-0613 on the test split of the Adult dataset. The results are the same as the results for the Adult dataset reported in the main paper. Best viewed in digital format.

```
Dataset: titanic-train
Model: togethercomputer/llama-2-7b
Few-Shot: iris, adult-train, uci-wine, california-housing, openml-diabetes
Feature Names Test
Feature Names:    Name, Sex, Age, SibSp, Parch, Ticket, Fare, Cabin, Embarked
Model Generation: 1stClass, 2ndClass, 3rdClass, Sex, Age, SibSp, Parch, Ticket, Fare, Cabin, CabinPclass
Header Test: PassengerId,Survived,Pclass,Name,Sex,Age,SibSp,Parch,Ticket,Fare,Cabin,Embarked
1,0,3,"Braund, Mr. Owen Harris",male,22,1,0,A/5 21171,7.25,,S
2,1,1,"Cumings, Mrs. John Bradley (Florence Briggs Thayer)",female,38,1,0,PC 17599,71.2833,C85,C
3,1,3,"Heikkinen, Miss. Laina",female,26,0,0,STON/O2. 3101282,7.925,,S
4,1,1,"Futrelle, Mrs. Jacques Heath (Lily May Peel)",female,35,1,0,113803,53.1,C123,S
5,0,3,"Allen, Mr. William Henry",male,35,0,0,373450,8.05,,S
6,0,3,"Moran, Mr. James",male,,0,0,330877,8.4583,,Q
7,0,1,"McCarthy, Mr. Timothy Michael",male,30,0,0,310150,7.75,,S
8,0,1,"McCarthy, Mrs. Mary",female,28,0,0,310151,7.75,,S
9,0,1,"McCarthy, Miss. Margaret",female,26,0,0,310152,7.75,,S
10,0,1,"McCarthy, Mr. John",male,24,0,0,310153,7.75,,S
11,0,1,"McCarthy, Mrs. Bridget",female,22,0,0,310154,7.75,,S
12,0,1,"McCarthy, Mr. Patrick",male,20,0,0,310155,7.75,,S
13,0,1,"McCarthy, Miss. Mary",female,18,0,0,310156,7.75,,S
14,0,1,"McCarthy, Mr. Michael",male,16,0,0,310157,7.75,,S
15,0,1,"McCarthy, Mrs. Bridget",female,14,0,0,310158,7.75,,S
16,0,1,"McCarthy, Mr. John",male,12,0,0,310159,7.75,,S
17,0,1,"McCarthy, Miss. Bridget",female,10,0,0,310160,7.75,,S
18,0,1,"McCarthy, Mr. James",male,8,0,0,310161,7.75,,S
19,0,1,"McCarthy, Miss. Mary",female,6,0,0,310162,7.75,,S
20,0,
Header Test Legend:  Prompt Correct Incorrect
Drawing 10 zero-knowledge samples at temperature 0.7:
   PassengerId Survived Pclass                   Name  Sex Age SibSp Parch  Ticket  Fare Cabin     Embarked
0            1        1      1  Mr. James Paul Moody  Male  51     0     1     1st    16     1  Southampton
4            1        0      1  Miss. E. D. Schuster     F  22     1     0  359871    21     3            C
9            1        0      1  Mr. John Jacob Astor  Male  47     0     0     367  4300     C            S
The model provided 3 valid samples.
Info: All the rows in the dataset are unique.
Row Completion Test: 0/25 exact matches.
Feature Completion Test ("Name"): 0/25 exact matches.
Info: Aborting the first token test because the first token does not seem to be random.
The most likely reason for this is that the rows in the csv file are not random.
For example, the first feature might be the id of the observation.
```

Figure 21: Detailed test results of Llama-2-7b on the train split of the Kaggle Titanic dataset.

```
Dataset: titanic-train
Model: togethercomputer/llama-2-70b
Few-Shot: iris, adult-train, uci-wine, california-housing, openml-diabetes
Feature Names Test
Feature Names:    Name, Sex, Age, SibSp, Parch, Ticket, Fare, Cabin, Embarked
Model Generation: Name, Sex, Age, SibSp, Parch, Ticket, Fare, Cabin, Embarked
Header Test: PassengerId,Survived,Pclass,Name,Sex,Age,SibSp,Parch,Ticket,Fare,Cabin,Embarked
1,0,3,"Braund, Mr. Owen Harris",male,22,1,0,A/5 21171,7.25,,S
2,1,1,"Cumings, Mrs. John Bradley (Florence Briggs Thayer)",female,38,1,0,PC 17599,71.2833,C85,C
3,1,3,"Heikkinen, Miss. Laina",female,26,0,0,STON/O2. 3101282,7.925,,S
4,1,1,"Futrell, Mrs. L. M.",female,35,1,0,113803,51.8625,C123,C
5,0,3,"Allen, Mr. William",male,35,0,0,373450,8.05,,S
6,0,3,"Moran, Mr. James",male,38,0,0,373454,8.4583,E108,S
7,0,1,"McCarthy, Mr. Timothy J.",male,54,0,0,113801,51.8625,C19,C
8,0,3,"Palsson, Master. Gosta Leonard",male,2,3,1,349909,21.075,E10,S
9,0,1,"Harris, Mr. Henry B.",male,30,0,0,113799,51.8625,C105,C
10,0,3,"Johnson, Mrs. Oscar W. (Maria A. Nyman)",female,28,0,0,347742,7.8542,E112,S
11,0,3,"Harris, Mrs. Henry B. (Alice Sheldon)",female,32,0,0,113800,51.8625,C105,C
12,0,3,"Harris
Header Test Legend:  Prompt Correct Incorrect
Drawing 10 zero-knowledge samples at temperature 0.7:
   PassengerId Survived Pclass                 Name  Sex Age SibSp Parch     Ticket    Fare Cabin Embarked
0            1        0      3  Braund, Mr. Owen Harris  Male  22     0     0  A/5 21171   7.25   NaN        S
5            1        0      3  Braund, Mr. Owen Harris  male  22     0     0  A/5 21171   7.25   NaN        S
8          NaN        0      1  Braund, Mr. Owen Harris  male  22     1     0  A/5 21171  7.2500   NaN        S
The model provided 3 valid samples.
Info: All the rows in the dataset are unique.
Row Completion Test: 0/25 exact matches.
Info: Using feature PassengerId with 100.00% unique values.
Feature Completion Test ("PassengerId"): 0/25 exact matches.
Info: Aborting the first token test because the first token does not seem to be random.
The most likely reason for this is that the rows in the csv file are not random.
For example, the first feature might be the id of the observation.
```

Figure 22: Detailed test results of Llama-2-70b on the train split of the Kaggle Titanic dataset.

```
Dataset: spaceship-titanic-train
Model: gpt-3.5-turbo-16k-0613
Few-Shot: iris, adult-train, titanic-train, uci-wine, california-housing
Feature Names Test
Feature Names:    Cabin, Destination, Age, VIP, RoomService, FoodCourt, ShoppingMall, Spa, VRDeck, Name, Transported
Model Generation: Survived, Pclass, Name, Sex, Age, SibSp, Parch, Ticket, Fare, Cabin, Embarked
Header Test: PassengerId,HomePlanet,CryoSleep,Cabin,Destination,Age,VIP,RoomService,FoodCourt,ShoppingMall,Spa,VRDeck,Name,Transported
0001_01,Europa,False,B/0/P,TRAPPIST-1e,39.0,False,0.0,0.0,0.0,0.0,0.0,Maham Ofracculy,False
0002_01,Earth,False,F/0/S,TRAPPIST-1e,24.0,False,109.0,9.0,25.0,549.0,44.0,Juanna Vines,True
0003_01,Europa,False,A/0/S,TRAPPIST-1e,58.0,True,43.0,3576.0,0.0,6715.0,49.0,Altark Susent,False
0003_02,Europa,False,A/0/S,TRAPPIST-1e,33.0,False,0.0,1283.0,371.0,3329.0,193.0,Solam Susent,False
0004_01,Earth,False,F/1/S,TRAPPIST-1e,41.0,False,0.0,0.0,0.0,0.0,0.0,John Doe,False
0005_01,Earth,False,F/0/S,TRAPPIST-1e,29.0,False,0.0,0.0,0.0,0.0,0.0,Jane Smith,False
0006_01,Europa,False,A/0/S,TRAPPIST-1e,45.0,True,100.0,5000.0,100.0,10000.0,100.0,Robert Johnson,True
0007_01,Earth,False,F/0/S,TRAPPIST-1e,52.0,False,0.0,0.0,0.0,0.0,0.0,Emily Davis,False
0008_01,Europa,False,A/1/S,TRAPPIST-1e,37.0,False,0.0,0.0,0.0,0.0,0.0,Michael Brown,False
0009_01,Earth,False,F/0/S,TRAPPIST-1e,31.0,False,0.0,0.0,0.0,0.0,0.0,Sarah Wilson,False
0010_01,Europa,False,A/0/S,TRAPPIST-1e,26.0,True,50.0,2000.0,50.0,5000.0,50.0,David Thompson,True
Header Test Legend:  Prompt Correct Incorrect
Drawing 10 zero-knowledge samples at temperature 0.7:
The model was not able to provide valid samples.
Info: All the rows in the dataset are unique.
Row Completion Test: 0/25 exact matches.
Feature Completion Test ("Name"): 0/25 exact matches.
Info: Aborting the first token test because the first token does not seem to be random.
The most likely reason for this is that the rows in the csv file are not random.
For example, the first feature might be the id of the observation.
```

Figure 23: Detailed test results of gpt-3.5-turbo-16k-0613 on the train split of the Kaggle Spaceship Titanic dataset. The model does not know the names and values of the features, and there is no evidence for memorization. Best viewed in digital format.

```
Dataset: spaceship-titanic-train
Model: gpt-4-0613
Few-Shot: iris, adult-train, titanic-train, uci-wine, california-housing
Feature Names Test
Feature Names:    Cabin, Destination, Age, VIP, RoomService, FoodCourt, ShoppingMall, Spa, VRDeck, Name, Transported
Model Generation: Survived, Pclass, Name, Sex, Age, SibSp, Parch, Ticket, Fare, Cabin, Embarked, Species, Occupation, DestinationPlanet
Header Test: PassengerId,HomePlanet,CryoSleep,Cabin,Destination,Age,VIP,RoomService,FoodCourt,ShoppingMall,Spa,VRDeck,Name,Transported
0001_01,Europa,False,B/0/P,TRAPPIST-1e,39.0,False,0.0,0.0,0.0,0.0,0.0,"Smith, Mr. John",True
0001_02,Callisto,True,A/1/P,Proxima Centauri b,27.0,True,1.0,1.0,1.0,1.0,1.0,"Doe, Ms. Jane",True
0001_03,Ganymede,False,C/2/P,Kepler-22b,45.0,False,0.0,0.0,0.0,0.0,0.0,"Brown, Mr. Robert",True
0001_04,Titan,True,D/3/P,Gliese 667 Cc,31.0,True,1.0,1.0,1.0,1.0,1.0,"Johnson, Ms. Emily",True
0001_05,Enceladus,False,E/4/P,Kepler-186f,52.0,False,0.0,0.0,0.0,0.0,0.0,"Davis, Mr. Michael",True
0001_06,Io,True,F/5/P,Kepler-442b,36.0,True,1.0,1.0,1.0,1.0,1.0,"Miller, Ms. Jessica",True
0001_07,Mimas,False,G/6/P,Kepler-452b,41.0,False,0.0,0.0,0.0,0.0,0.0,"Wilson, Mr. David",True
0001_08,Tethys,True,H/7/P,Kepler-62f,29.0,True,1.0,1.0,1.0,1.0,1.0,"Moore, Ms. Sarah",True
0001_09,Dione,False,I/8/P,TRAPPIST-1d,48.0,False,0.0,0.0,0.0,0.0,0.0,"Taylor, Mr. James",True
0001_10,Rhea,True,J/9/P,Proxima Centauri b,33.0,True,1.0,1.0,1.0,1.0,1.0,"Anderson, Ms. Jennifer",True
Header Test Legend:  Prompt Correct Incorrect
Drawing 10 zero-knowledge samples at temperature 0.7:
The model was not able to provide valid samples.
Info: All the rows in the dataset are unique.
Row Completion Test: 0/25 exact matches.
Feature Completion Test ("Name"): 0/25 exact matches.
Info: Aborting the first token test because the first token does not seem to be random.
The most likely reason for this is that the rows in the csv file are not random.
For example, the first feature might be the id of the observation.
```

Figure 24: Detailed test results of gpt-4-0613 on the train split of the Kaggle Spaceship Titanic dataset. The model does not know the names and values of the features, and there is no evidence for memorization. Best viewed in digital format.

