# OpenReview forum: "Elephants Never Forget: Testing Language Models for Memorization of Tabular Data"
_ICLR.cc/2024/Conference — Submitted to ICLR 2024_

### Official Review · Reviewer_YJg3 · 2023-10-21

**Soundness:** 2 fair
**Presentation:** 1 poor
**Contribution:** 2 fair
**Rating:** 3
**Confidence:** 4

**Summary:**

This paper aims at inspecting data contamination which happens when training a large language model. Specifically, they inspect whether some tabular datasets were used to train the language models. For this purpose, they propose several approaches to testing the language models.
- Testing for Knowledge and Learning:
    - Meta data: Testing whether the model can predict the name of the fields in the datasets.
    - Conditional completion: Testing whether the model can predict the value of a feature of a sample based on some other features.
    - Unconditional zero-knowledge samples: Testing whether the model produces the statistics of the features in a datasets.
- Testing for memorization:
    - Testing whether the model can generate rows based on the previous rows (either the first a few rows or a few rows starting from a random line) in the dataset csv file.
    - Testing whether the model can generate the correct value of a feature based on the value of the other features. Here the value is “unique”.
    - Testing whether the model can generate the first token of the next row.

Finally, they compare the language models’ performance on several datasets with some baseline models’ performance, and conclude that some models’ performance can be attributed to their memorization of the datasets during pretraining.

**Strengths:**

- They propose several methods to test whether the language models were trained with those datasets.
- The idea of comparing the distribution generated by the model and the distribution in the datasets (Sec 3.3) is novel and interesting.
- The claim that they show that some language models are pretrained with some tabular datasets is somewhat convincing and interesting.

**Weaknesses:**

Main concerns:

1. This work does not provide strong evidence supporting the validity of their proposed approaches. I think one main takeaway of this paper is that some models are pretrained with some datasets, so their performance is not indicative of. But this takeaway is based on the validity of their proposed approaches. I think the authors need to address this more.
2. I can’t understand the purpose of having these many different testing approaches, probably because the structure of this paper is hard to follow. The authors propose many approaches, some of them are interesting, but they do not provide a holistic interpretation of the results from these many approaches.
3. The descriptions of the testing approaches are vague and not rigorous. Writing down the testing approaches with simple math equations could help. For example, in page 6, I can’t understand what it means by “we can perform a t-test between the similarity of model completions with actual vs. random rows.
4. Knowledge, learning, memorization should be defined more specifically.
5. The authors (claims to) show data contamination exists in some datasets. However, I am not sure whether those datasets are commonly used to benchmark the language model. Thus I am not sure whether the findings are important (if they are valid).

More specific (writing) issues:


2. The second paragraph in Sec 3.2: Here 4 possible causes are provided, but I don’t see how they are discussed in the following experimental designs.
3. The last sentence in page 5: “Empirically, we find … a very intuitive test …”. I don’t understand how your empirical results support this.
4. Page 8: “It might be that this learning task is relatively simple, that our memorization test are not sensitive enough”. I can’t understand why it is the case.


Grammar:

There are many grammar errors. I suggest that the authors do some proofreading.

**Questions:**

1. Figure 1: I suggest to use bar charts is more reasonable because your x-axis is not continuous.

---

> ### Author Response · Authors · 2023-11-22
> **Author Response to Reviewer YJg3**
>
> We thank the reviewer for the detailed review and many helpful comments.
>
> - *“This work does not provide strong evidence supporting the validity of their proposed approaches.”*
>
> With regard to the validity of our tests, please see the global comment ‘The validity and relevance of our tests’.
>
> In addition, note that we have been careful to use (1) datasets that are public and overwhelmingly likely to have been used in pre-training, and (2) datasets that are very unlikely or impossible to have been used in pre-training.  There is a clear difference in the results for these two different kinds of datasets.  We also intentionally include FICO as a dataset where there is significant public discussion of the data and some of its properties, but where the data table itself has not been made public. As might be expected, on this dataset our tests detect prior experience but fail to detect verbatim memorization.
>
> - *“I can’t understand the purpose of having these many different testing approaches”*
>
> We acknowledge that the flow of the paper can be overwhelming due to the many different proposed approaches. Under the limitations we impose upon our testing framework, we believe it is impossible to design a single test that will work perfectly under all possible circumstances. Our design philosophy therefore is to design a battery of tests that offer empirical evidence of memorization in concert. Under stricter assumptions, a smaller number of tests may likely be sufficient, but making stronger assumptions would compromise the utility of our tests by working either only with fewer LLMs or fewer tabular datasets. Overall, we believe that the proposed tests allow us to provide a nuanced picture of different types of pre-training contamination.
>
> - *“The descriptions of the testing approaches are vague and not rigorous.”*
>
> We agree that there is room for improvement and will rewrite the sections describing the different tests. For details on the different tests, please also see Supplement Section E which gives the prompt structures of all the tests, and the revised Supplement Section F which provides the test results in a more explicit form.
>
> - *“I am not sure whether those datasets are commonly used to benchmark the language model. Thus I am not sure whether the findings are important”*
>
> Yes, the datasets in our paper have been used to benchmark language models. Please see the papers referenced under ‘The relevance of our work’ in the global comment.

---

### Official Review · Reviewer_aXk7 · 2023-10-22

**Soundness:** 2 fair
**Presentation:** 3 good
**Contribution:** 2 fair
**Rating:** 3
**Confidence:** 4

**Summary:**

The paper presents a discussion on data contamination and memorization in large language models concerning tabular data. The authors propose multiple methods to examine whether an LLM has memorized specific tabular datasets during training. The paper also proposes methods to examine knowledge, learning, and memorization separately and discuss the distinction between them. Finally, the paper analyzes the influence of learning and memorization on the performance of downstream tasks, and advocates checking memorization as a crucial step in evaluating LLM on tasks with tabular data.

**Strengths:**

The paper presents several novel methods to evaluate memorization of tabular data in LLMs, and evaluation results on a series of datasets correlate well with the publication time and availability of the data, confirming the effectiveness of the proposed methods in identifying memorization. The different evaluation methods also complement each other, elucidating the different aspects of memorization of tabular data.

The paper is overall well-written and very easy to read, the visualizations present the main findings nicely.

The contamination and memorization of training data by LLMs is a critical issue. The findings provoke essential discussions on the evaluation of LLMs on tabular data, which is likely to become more relevant given the rising usage of LLMs in diverse tasks.

The introduced tools and code potentially provide easy and accessible ways to evaluate memorization of tabular data, reusable in future research.

**Weaknesses:**

Some important details in the experiment design may be missing or incomplete:

* Evaluation metric for knowledge, learning, and memorization is unclear. In Table 1, the evaluation results are categorized into three categories (✓,X, and ?), but the metric for the categorization is not given. It is probably a better idea to show the raw values (e.g., accuracy) than using categories to give the reader a direct comprehension of the degree of memorization on each dataset. Notations such as "✓" could be misleading as it may be confused as perfect memorization.

  The appendix gives raw accuracies for Row Completion Test, Feature Completion Test, and First Token Test, why raw accuracies for Feature Names, Feature Values, and Header Test are not provided as well?

* The differentiation between learning and memorization is not clear: the authors use feature distributions to examine learning, but memorization can also result in a high similarity of the generated data's feature distributions to the original data. Learning is defined as the model's ability to perform tasks in the current paper, but task performance is heavily affected by memorization and may fail to reflect true learning. Even with considerable discussion, the paper does not seem to arrive at a conclusion about how learning can be clearly assessed.

* Evalulation of memorization needs to take the nature of data fields into consideration. Some data fields in the tabular dataset are considerably harder to memorize verbatim or to predict exactly (such as measurement values) than other simpler fields (categorical values such as sex, occupation, nation). For numerical values, it may be more reasonable to measure the relative distance from the predicted value to the true value than using exact match (perhaps in a similar vein as the "first token test" in the paper but more principled).

  Under the current evaluation protocol, it is likely that datasets containing more easy fields are more likely to be judged as memorized. To compare the degree of memorization across datasets, it seems necessary to perform some kind of "normalization" before measuring memorization, for example, selecting a fixed number of categorical and numerical fields from each dataset. Results in Table 3 could suffer from this limitation as well.

* Evaluation of memorization needs to be evaluated separately for the training and test split. It may be possible that the training sets are memorized more than the test set due to more exposure on the internet. Memorizing the test set definitely compromises evaluation, but memorizing the training set may not always compromise evaluation.

* Connection between memorization and downstream performance is not reliably established. The main observation from Section 5 is that for datasets with a high degree of memorization, LLM performs better than decision tree and logistic regression, while for datasets with a low degree of memorization the reverse is true. Such observation alone may not be sufficient to conclude that memorization compromises evaluation, because there is no evidence that LLM cannot perform better than decision tree and logistic regression under no memorization. It would be much better to solicit new test sets for the tasks to use in evaluation, which can be used to show exactly how much performance gap is caused by memorization. In case finding new examples is difficult, perhaps one can modify the values of the fields known to be irrelevant to the label in existing examples, and that may break the reliance on memorization in LLMs.

Some main conclusions of the paper are compromised because of the above limitations:

* "We emphasize the importance of verifying data contamination before applying LLMs": the implication of data contamination is not reliably demonstrated in Section 5.

  Also, from the current discussion, it is not very clear how to interpret the test results on knowledge, learning, and memorization together. For example, if knowledge and learning show positive results and memorization show negative results, should we conclude that there is data contamination or not? And could the performance on downstream tasks be trusted in this situation?

  It can be argued that knowledge and learning will not directly compromise evaluation on downstream tasks, so there may not be as much need to evaluate them compared to memorization. I would suggest allocating more space in the paper for extended experiments and discussions on memorization, which is the ultimate reason why people are concerned about data contamination.

* "... and propose practical methods to do so": the proposed method verifies memorization of data, but does not give a definite metric to judge when memorization is severe enough to compromise evaluation.

* "We offer a principled distinction between learning and memorization in LLMs": the distinction is not given clearly enough. One can tell whether there is memorization from the proposed test, but it is not clear how to tell whether learning exists (especially when memorization is present).

**Questions:**

What is the difference in the remembering behavior of LLM between tabular data and non-tabular data? The question may help strengthen the original contribution of the paper.

In the beginning of the discussion section, the use of the term "representation learning" may be confusing to some people. Representation learning usually refers to the process of learning useful features from raw data (wikipedia), which does not include memorization by definition.

---

> ### Author Response · Authors · 2023-11-22
> **Author Response to Reviewer aXk7 (Part 1)**
>
> We thank the reviewer for the detailed review and many helpful comments.
>
> - *“raw accuracies for Feature Names, Feature Values, and Header Test are not provided ”*
>
> We empirically found that the language models either (1) correctly list all the feature names, or (2) completely fail to list the feature names. The same holds true for the Feature Values test (third row of Table 3) and also for the header test. This is why we decided to depict the test results qualitatively. We agree that we did not make this clear enough in the paper. The revised Supplement Section F now depicts detailed results of the different tests.
>
> - *“the authors use feature distributions to examine learning, but memorization can also result in a high similarity of the generated data's feature distributions to the original data. Learning is defined as the model's ability to perform tasks in the current paper, but task performance is heavily affected by memorization and may fail to reflect true learning. Even with considerable discussion, the paper does not seem to arrive at a conclusion about how learning can be clearly assessed.”*
>
> Great point. We find it important to introduce the learning/memorization distinction to make clear that there are many different tasks that the model might be able to perform with the data that are not indicative of memorization (see also ‘The distinction between Knowledge, Learning and Memorization’ above). We are reasonably confident to have identified cases where the LLMs have learned, but not memorized. The reviewer asks whether we are able to test for learning under the assumption that there is memorization. This is a very interesting question (and also a question that is hard to answer). With the tests that we propose in this paper, we do not claim to be able to do this.
>
> - *“Evaluation of memorization needs to take the nature of data fields into consideration. [...] For numerical values, it may be more reasonable to measure the relative distance from the predicted value to the true value than using exact match (perhaps in a similar vein as the "first token test" in the paper but more principled). Under the current evaluation protocol, it is likely that datasets containing more easy fields are more likely to be judged as memorized. To compare the degree of memorization across datasets, it seems necessary to perform some kind of "normalization" before measuring memorization, for example, selecting a fixed number of categorical and numerical fields from each dataset. Results in Table 3 could suffer from this limitation as well.”*
>
> This is an interesting observation, but we tend to disagree. In fact, it is precisely the exact matches between rare numerical values in the data and the generations by the LLM that allow us to develop valid tests for memorization. The fact that tabular datasets often contain such values (as opposed to free-form text) is what makes tabular data such an interesting testbed for memorization (and this is also why the results in Table 3 are valid). For a dataset that only contains easy fields such as categorical variables with few different values, it would be very hard to distinguish a powerful learner and memorization. Measuring the relative distance from the predictive value to the true value could result in a test for learning, but would be unlikely to result in a valid test for memorization.
>
> - *“Evaluation of memorization needs to be evaluated separately for the training and test split. ”*
>
> Very good point. The revised Supplement Section F now contains separate evaluations for the datasets where separate train and tests splits are available. We do not find any significant differences.
>
> - *“Connection between memorization and downstream performance is not reliably established. [...] It would be much better to solicit new test sets for the tasks to use in evaluation, which can be used to show exactly how much performance gap is caused by memorization.”*
>
> The SOTA accuracy for Kaggle Titanic using ensembling is approximately 0.85 (see https://www.kaggle.com/code/pliptor/how-am-i-doing-with-my-score/report). GPT-4 achieves an accuracy of 0.98. We think that this is fairly indicative of memorization. We agree, however, that the approach in Section 5 is only a first step, and that one could think of more advanced evaluation strategies (such as the one suggested by the reviewer). In general, however, there are many possible downstream tasks, and it is not possible for tests of memorization and prior exposure to consider (or even be relevant) for all of these.  Because of this, we believe it is important to have tests that are also reasonably task-agnostic.

---

> > ### Author Response · Authors · 2023-11-22
> > **Author Response to Reviewer aXk7 (Part 2)**
> >
> > - *“if knowledge and learning show positive results and memorization show negative results, should we conclude that there is data contamination or not? And could the performance on downstream tasks be trusted in this situation?”*
> >
> > The setting that we are interested in here is when a paper proposes a new prompting or fine-tuning strategy, and then demonstrates that this strategy leads to improved performance on some datasets. If the model already shows positive results for learning on these datasets, then this raises the question of whether the results are really due to the new prompting/fine-tuning strategy, or reliant on the fact that the LLM has already learned about the dataset during pre-training. Even when testing does not show memorization, prior learning can significantly bias the results that are obtained in experiments by allowing the LLM to perform advanced tasks with the data distribution that it would not have been able to perform on a novel dataset without prior exposure. Papers that study tabular learning with LLMs are, in most cases, purporting to show results that would be obtained without prior data contamination (see also ‘The relevance of our work’ above).
> >
> > - *“What is the difference in the remembering behavior of LLM between tabular data and non-tabular data?”*
> >
> > This is a very interesting and important question. In our view, it is much more difficult to evaluate prior contamination of LLMs on textual data. This is because textual data lacks the specific structure found in tabular datasets - prior experience with textual data can come in many different forms, and it is unclear how to design canonical tests. Of course, if an LLM exactly reproduces a long string of text or a long poem or song lyric, this is almost certainly a sign of prior exposure and memorization. But if the LLM fails to reproduce the long string exactly (or even approximately) there remain many plausible ways in which the LLM might still have seen or memorized the relevant information. For example, the LLM might have memorized the poem or song lyric in a different language, or in a reworded or otherwise re-formatted version (while similar problems can in principle also occur for tabular datasets, we believe that these problems are much exacerbated for natural language).

---

### Official Review · Reviewer_ELKr · 2023-10-27

**Soundness:** 4 excellent
**Presentation:** 4 excellent
**Contribution:** 3 good
**Rating:** 8
**Confidence:** 4

**Summary:**

LLMs are increasingly being applied to various types of data, including tabular data. Since, at the moment, the most advanced LLMs are essentially black-boxes w/ restricted APIs with little details available about their training data, it is hard to tell a priori whether the tabular data was leaked into the training and whether the models have memorized it.
This paper proposes four tests that probe an LLM for the training data contamination and estimate the degree of the contamination (“knowledge”, “learning”, “memorization”).

In the experimental study, the paper focuses on ChatGPT 3.5 & 4 and 10 tabular datasets, which have a high chance of being in the training data obtained by crawling the Internet (Iris, Kaggle Titanic, …).

First, the authors show that both LLMs have memorized basic meta-data from the datasets. Next, the LLMs are probed for an ability to reproduce a dataset example, conditioned on a part of its features. As an example, on the Adult Income dataset, the LLMs completed EduNum feature significantly better than a marginal distribution baseline. Further, the authors propose “zero-knowledge” prompting technique where the model is prompted to sample samples from a dataset (unconditionally or conditionally), provided samples from other datasets. Using that approach, the authors show that the models often can reproduce the distribution of the data in some datasets (approximately). Finally, the models are probed to reproduce parts of the datasets verbatim; for some datasets that happens extremely often.

Additionally, S5 provides a comparative analysis of the LLMs performing few-shot classification tasks on a subset of datasets, in comparison to standard ML baselines. It turns out that the LLMs have marked drop in performance on some datasets that are likely absent in training (Pneumonia & Spaceship Titanic); at the same time very high performance on datasets that are likely to be memorized (Kaggle Titanic).

**Strengths:**

* I believe this work (a) raises an important overlooked question, (b) addresses it, (c) by proposing an original technique. I particularly like the zero-shot prompting technique that allows sampling from a dataset w/o leaking information in the prompt.
* The paper disentangles a few levels of training data contamination and comprehensively tests for those.
* The paper showcases the potential impact of the contamination on the downstream comparisons, hence proving a strong motivation to the work.
* The text and the story are clear.
* The code is made public.

**Weaknesses:**

* The paper only studies ChatGPT-3.5 and 4. Those are very likely to be strongly correlated in terms of the data used, which harms the representativeness of the study.
* As there is no ground-truth knowledge on whether a particular dataset was seen at training, it is impossible to strictly verify the findings. Including an LM trained on a known dataset would allow us to verify the used methods.
* Another related issue: the work is mostly relevant when we consider closed-data models w/ a black-box API access. This scenario reflects a dominant situation at the moment, but it is not given that this will not/should not change.

Minor:
* Table 1 is mentioned on page 3, yet only appears on page 6. Is there a way to bring it closer?
* Would it make sense to consider swapping sections 5 and 6? I feel the S6 is more connected to the S3-4 than S5.

**Questions:**

I wonder if authors would be willing to address the first two points in ‘weaknesses’.

---

> ### Author Response · Authors · 2023-11-22
> **Author Response to Reviewer ELKr**
>
> We thank the reviewer for the detailed review and appreciation of our paper. We also thank the reviewer for the minor comments that we will incorporate into the paper.
>
> - *“The paper only studies ChatGPT-3.5 and 4. ”*
>
> We have now extended our code and prompts to work with arbitrary language models (some adaptation of prompts is required between base models and chat models, and the capabilities of chat models to follow instructions vary). The revised Supplement Section F contains results with Llama2-7b and Llama2-70b.
>
> - *“Including an LM trained on a known dataset would allow us to verify the used methods.”*
>
> We agree that this is an interesting direction in which our work could be extended. Unfortunately, the LLMs for which the training data and schedule are transparent are rather small (<100b Parameters). At the same time, memorization is known to occur especially for very large models. This is evident, for example, in Supplement Table 4, which shows that GPT-4 has memorized significantly more than GPT-3.5.
> It is also evident in the Figures in the revised Supplement Section F, where it can be seen that Llama2-7b is the model that has memorized the least.
>
> - *“the work is mostly relevant when we consider closed-data models w/ a black-box API access. “*
>
> We completely agree. We also believe that this is a very important and relevant scenario (see also ‘The validity and relevance of our tests’ above).

---

### Official Review · Reviewer_8936 · 2023-11-01

**Soundness:** 2 fair
**Presentation:** 2 fair
**Contribution:** 2 fair
**Rating:** 5
**Confidence:** 3

**Summary:**

This paper specifically targets the issue of contamination in training sets when evaluating LLMs on tasks with tabular data.
Compared to previous work on LLMs for tabular data, the authors propose methods to test the LLM for memorization (in addition to the dimensions "knowledge" and "learning").
These novel tests help to better analyze and understand the performance on downstream tasks, such as deciding if the data has been seen in training or not.

**Strengths:**

* LLMs are pervasive currently, and it's important to understand and control their behavior. The authors emphasize the importance of verifying data contamination before applying LLM.
* Their setup based on tabular data is an elegant way to test “knowledge”, “learning”, and “memorization” of an LLM.
* Moreover, they assume only blackbox API access, without assuming access to the probability distributioin over tokens or the ability to re-train the model.
* Release of an open-source tool that can perform various tests for memorization.

**Weaknesses:**

* My main point of criticism is that the paper feels a bit like a collection of remarkable examples and the analysis largely confirms known concerns/behavior of LLMs.
* "we also identify a regime where the language model reproduces important statistics of the data, but fails to reproduce the dataset verbatim": It's not clear to me what this statement means. See also Question 1 below.

* Just echoing the authors: "A limitation of our work is that we do not have access to the training data of GPT-3.5 and GPT4." I.e., the interpretation of results often remains speculative.
* Figure 3: Why are some results with gpt-3.5 and some with gpt-4?
* Typo: "two publicly available dataset that are highly memorized"
* Typo: "UCI repository athttps://"

**Questions:**

1. "​​An important result of our investigation is to identify a regime where the LLM has seen the data during training and is able to perform complex tasks with the data": Don't LLMs behave as expected on some data and not on other? How does this work help to control the behavior of LLMs?

---

> ### Author Response · Authors · 2023-11-22
> **Author Response to Reviewer 8936**
>
> We thank the reviewer for the detailed review and many helpful comments.
>
> - *“My main point of criticism is that the paper feels a bit like a collection of remarkable examples and the analysis largely confirms known concerns/behavior of LLMs. ”*
>
> We agree that dataset contamination is a general concern in LLMs, and that it is widely known that LLMs can memorize their training data. Despite this, many papers at the top ML conferences evaluate LLMs on datasets that potentially are part of the LLM’s pre-training corpus without performing any systematic tests for memorization (see ‘The relevance of our work’ above) . We suspect that a main reason for this is that while contamination is a known concern, there are few tools available that allow researchers to systematically test for memorization and other forms of prior contamination within the timeframe of a single research project. With this paper, we attempt to develop testing approaches that will allow future research to address this question in a systematic manner.
>
> - *“we also identify a regime where the language model reproduces important statistics of the data, but fails to reproduce the dataset verbatim”*
>
> Here we are referring to the behavior of GPT-3.5 on the California Housing dataset. On this dataset, the samples provided by the model match the correlations in the training data (Figure 3). At the same time, the average n-gram match between the samples and the training data is only 3.1/10 (Table 3) and there is no evidence for verbatim memorization except for the header of the dataset (Table 1).
>
> - *“the interpretation of results often remains speculative.”*
>
> We would not call our results speculative. We don’t think that there is any plausible way in which the model could reproduce tabular datasets verbatim (Table 1) or with precisely matching statistics (Figure 3) without having seen those datasets during pre-training.
>
> - *Figure 3: “Why are some results with gpt-3.5 and some with gpt-4?”*
>
> Figure 3 depicts selected examples. The full results, for both gpt-3.5 and gpt-4, are in Supplement D.
>
> - *“"​​An important result of our investigation is to identify a regime where the LLM has seen the data during training and is able to perform complex tasks with the data": Don't LLMs behave as expected on some data and not on other? How does this work help to control the behavior of LLMs?”*
>
> We don’t know what the reviewer considers the expected behavior of LLMs to be. We would interpret our results as showing that there is a fair degree of variation in LLM behavior, both across different models, and also across datasets for a single model.

---

> > ### Comment · Reviewer_8936 · 2023-11-23
> >
> > Thanks for your clarifications.
> >
> > > “​​An important result of our investigation is to identify a regime where the LLM has seen the data during training and is able to perform complex tasks with the data."
> >
> > I'm still not sure if I understand this statement: Is it (a) You identified "knobs" to control the behaviour of the LLM and tested it on a number of data sets or rather (b) You use an existing LLM and find data sets that exhibit the ability to perform complex tasks with the data?

---

### Author Response · Authors · 2023-11-22
**Author Rebuttal**

We thank the reviewers for taking the time to review our paper and for their many helpful comments.

**The relevance of our work:**

Testing for memorization of tabular datasets in LLMs is of practical relevance due to the increasing number of works that employ LLMs for tasks with tabular data. We encourage reviewers who wonder about the relevance of our work to take a quick look at the following papers, all of which employ LLMs for tasks with tabular data without testing for any form of contamination:

[1] Hollman et al., *Large Language Models for Automated Data Science: Introducing CAAFE for Context-Aware Automated Feature Engineering*, NeurIPS 2023

[2] Borisov et al, *Language models are realistic tabular data generators*, ICLR 2023

[3] Hegselmann et al., Tabllm: *Few-shot classification of tabular data with large language models*, AISTATS 2023

The main objective of our paper is to call attention to this important, but underappreciated, issue, and to propose tests that can detect memorization, learning, and other forms of prior familiarity because any of these can bias the results of experiments performed with tabular data.

**The validity and relevance of our tests:**

The validity of our memorization tests is similar to the validity of the tests in Carlini et. al (2021). We exploit the particular structure of tabular datasets which contain specific realizations of random variables (‘canaries’) that – in their combination –  are statistically unlikely to be generated even by the Bayes predictor.

Our main objective is to propose tests that will be useful to practitioners under the strict limitations of the most popular publicly available LLMs, like GPT-4, where we can neither access the training data nor the probability distribution of tokens (‘logprobs’) during completion. Under these restrictions, the tests that we can perform are not perfect, and no one test can detect all of the forms of contamination that might occur.  Nonetheless, we believe the tests we introduce are more powerful and systematic than other tests in the literature. In lieu of being able to confirm our results by having access to the training data for models like GPT-4, we run experiments on datasets that have been widely used and which almost certainly have been memorized, and datasets that we are confident have not been released publicly and which therefore cannot have been memorized. We believe that a battery of tests analyzed in concert can provide useful empirical evidence to practitioners. A main advantage of our tests is that they can all be performed automatically (see the additional Figures in the revised Supplement Section F).


**The distinction between Knowledge, Learning, and Memorization:**

We find it important to distinguish between knowledge, learning, and memorization to make it clear that there are forms of bias in LLM evaluation that are not due to verbatim memorization. All of these mechanisms can plausibly lead to optimistically good performance on tasks when tested with data the LLM has been exposed to during pre-training. We provide tests that identify memorization in the presence of learning but do not claim to provide tests that identify learning in the presence of memorization.

---

### Meta-Review · Area_Chair_aAJa · 2023-12-10

**Metareview:**

The paper introduces for testing whether LLMs have been trained on specific tabular data, which might indicate contamination of certain test sets. Reviewers agree that understanding contamination is important, and it is great that the proposed methods work with only black box access to the language model. Better tooling for measuring contamination would aid in interpreting LLMs, given that training data details are rarely disclosed. However, the reviewers identified several important limitations: (1) Because we don't know what GPT3.5/4 were trained on, we can't know if the method is making the correct predictions - although I agree it seems likely, it would be nice to test this with LLMs with known training data. (2) The distinctions between knowledge, learning, memorization are not clearly enough defined. (3) While these datasets have been used to evaluate LLMs, they are not the most commonly used tasks, limiting the impact of the approach. Overall, I think this paper needs a bit more work before it is ready for acceptance.

**Justification For Why Not Higher Score:**

See metareview, but I think this is not likely to be a high impact paper without a bit more work.

**Justification For Why Not Lower Score:**

N/A

---

### Decision · Program_Chairs · 2024-01-16

Reject